# Mosquito immune cells enhance dengue and Zika virus infection in *Aedes aegypti*

David R. Hall[1,2,5], Rebecca M. Johnson[3,5], Hyeogsun Kwon [2,5],
Zannatul Ferdous[3], S. Viridiana Laredo-Tiscareño[4], Bradley J. Blitvich[4],
Doug E. Brackney [3] & Ryan C. Smith [2] ✉

Mosquito-borne arboviruses cause more than 400 million annual infections, yet despite their public health importance, the mechanisms by which arboviruses infect and disseminate in the mosquito host are not well understood. Here, we provide evidence that dengue virus and Zika virus actively infect *Aedes aegypti* hemocytes and demonstrate, through phagocyte depletion, that hemocytes facilitate virus infection to peripheral tissues including the ovaries and salivary glands. Adoptive transfer experiments further reveal that virus-infected hemocytes efficiently confer virus infection to naïve recipient mosquitoes. Together, these data support a model of arbovirus dissemination where infected hemocytes enhance virus infection of mosquito tissues required for transmission, which parallels vertebrate systems where immune cell populations promote virus dissemination. This study significantly advances our understanding of virus infection dynamics in the mosquito host and highlights potential conserved roles of immune cells in arbovirus infection across vertebrate and invertebrate systems.

In recent decades, climate change and globalization have driven the expansion of *Aedes albopictus* and *Aedes aegypti* from their native tropical and subtropical habitats to more temperate environments across the globe, such that they are now present on every continent except Antarctica[1–3]. As the primary vectors of dengue (DENV), Zika (ZIKV), and many other arboviral diseases, the expansion of these mosquito species has increased the incidence of DENV by 30-fold in the last 50 years, reaching an estimated 100 million clinical cases per year[4]. Likewise, arboviral outbreaks, such as the emergence of ZIKV in the Americas in 2015 and 2016, underscore the increasing risk and public health threat presented by these mosquito species and the viruses they transmit[5]. With the risk of arbovirus transmission likely to continue to increase and further expand into new areas in the future[6–8], understanding the key factors that influence arbovirus transmission by the mosquito host is of significant importance.

After entering the mosquito through an infectious blood meal, viruses must overcome multiple tissue barriers in the mosquito host in order to reach the salivary glands and be transmitted to a new host[9–11]. While virus uptake and replication in the midgut are an essential first step to mosquito infection, the manner by which the virus disseminates from the midgut to other mosquito tissues remains poorly understood. Several potential routes of virus escape into the hemolymph and secondary tissues have been suggested, including routes through the tracheal system[12,13], damage to the midgut basal lamina[14–17], and infection of the visceral muscles[18]. Although each of these potential routes, or combinations thereof, may contribute to virus dissemination, recent evidence suggests that the midgut basal lamina may be the most significant barrier. Several studies have shown that damage to the basal lamina resulting from an additional blood meal significantly enhances virus dissemination, reducing the extrinsic incubation period[14,17,19].

[1]Interdepartmental Program in Genetics and Genomics, Iowa State University, Ames, IA, USA. [2]Department of Plant Pathology, Entomology and Microbiology, Iowa State University, Ames, IA, USA. [3]Center for Vector-Borne and Zoonotic Diseases, Department of Entomology, The Connecticut Agricultural Experiment Station, New Haven, CT, USA. [4]Department of Veterinary Microbiology and Preventative Medicine, Iowa State University, Ames, IA, USA. [5]These authors contributed equally: David R. Hall, Rebecca M. Johnson, Hyeogsun Kwon. ✉e-mail: smithr@iastate.edu

Mosquito immune cells, known as hemocytes, serve as important immune sentinels that circulate in the hemolymph and have integral roles in pathogen recognition, immune signaling, and wound healing[20,21]. Hemocytes are found either in circulation or attached to different mosquito tissues as sessile cells[22] and can readily become infected by virus[23–27]. As a result, it has been suggested that hemocytes may serve as an additional tropism for virus replication in the hemolymph[25]. While evidence indicates that mosquito and *Drosophila* hemocytes contribute to antiviral defenses[26,28–31], the potential that virus-infected hemocytes may also promote virus infection in the mosquito host has not yet been explored.

While previous studies in vertebrates have demonstrated the integral roles of immune cells in DENV and ZIKV infection, dissemination, and pathogenesis[32–38], similar studies of mosquito immune cells in arbovirus infection have been constrained by the lack of genetic tools available to manipulate these hemocyte populations. Therefore, previous studies in mosquitoes have been largely observational[23–25,27] or have relied on methods to impair phagocytic function[26]. To overcome many of these limitations, we have pioneered the use of clodronate liposomes to deplete phagocytic immune cells across arthropod systems[39–41]. Herein, we similarly utilize clodronate liposomes to deplete phagocytic hemocytes in *Ae. aegypti* to determine their respective role in DENV and ZIKV infection. Through our experiments, we demonstrate that phagocytic granulocytes are the hemocyte subtype predominantly infected by arboviruses and that their depletion attenuates virus dissemination. Additional transfer experiments of either acellular hemolymph or phagocytic granulocytes suggest that granulocytes are the primary hemolymph component required to transfer a virus infection. Together, these findings demonstrate the importance of hemocytes as a tropism for virus infection that enhances virus dissemination in the mosquito host.

## Results

### Granulocyte depletion does not influence midgut DENV or ZIKV titers

Mosquito hemocytes have integral roles in innate immunity, yet their roles in arbovirus infection have not been adequately addressed. To approach this question and to determine the influence of granulocytes on midgut virus infection, we employed recently established methods of chemical ablation using clodronate liposomes to selectively deplete phagocytic granulocytes[39–42]. As previously described[40], *Ae. aegypti* mosquitoes were injected with clodronate liposomes (CLD) to deplete granulocyte populations or with "empty" control (LP) liposomes. The effects of CLD treatment were long lasting, with a significant reduction in the percentage of granulocytes when evaluated using a hemocytometer for more than 10 days post-blood feeding (11 days post-injection, dpi) (Supplementary Fig. S1a). Of note, the percentage of granulocytes decreased over time in the LP-treated mosquitoes (Supplementary Fig. S1b), potentially due to an age-related decline in immune function[43]. Despite this, CLD treatment resulted in a decrease in the percentage of granulocytes that persisted over the entire time examined, such that granulocytes never recovered following CLD treatment (Supplementary Fig. S1a). This suggests that phagocyte depletion is efficient for at least 10 days after injection.

To determine the impact of phagocyte depletion on virus infection, mosquitoes were treated with LP or CLD, then challenged with DENV or ZIKV via a blood meal delivered using an artificial membrane system (Fig. 1a). Infection outcomes from individual experiments (Supplementary Fig. S2) were pooled to assess the effects of CLD treatment in dissected midgut samples at 7 dpi for DENV (Fig. 1b) or ZIKV (Fig. 1c) viral loads as determined by qRT-PCR. Following CLD treatment, no differences in DENV midgut titers or midgut infection prevalence were observed between LP- or CLD-injected groups (Fig. 1b, Supplementary Fig. S2). In addition, LP or CLD treatment similarly had no effect on midgut ZIKV titers, although our data display a significant

decrease in pooled infection prevalence in CLD-treated mosquitoes (Fig. 1c), consistent with trends from individual experiments (Supplementary Fig. S2). These results suggest that phagocytic granulocyte depletion does not have a significant impact on the intensity of midgut flavivirus infection, although significant differences in ZIKV infection prevalence indicate that granulocytes may affect other yet undescribed aspects of midgut physiology that influence the susceptibility to infection in specific virus-vector combinations.

To rule out the possibility that CLD treatment could interfere with virus infection and replication, we examined DENV and ZIKV infection in vitro in the presence of increasing concentrations of clodronate disodium salt (Fig. 1d), the encapsulated component of clodronate liposomes[44]. When examined in C6/36 cells, clodronate treatment had no effect on DENV (Fig. 1e) or ZIKV (Fig. 1f) infection and replication, suggesting that any phenotypes associated with CLD treatment are associated with phagocyte depletion.

### Granulocyte depletion does not impact mosquito survival following virus infection

Following the depletion of phagocytic granulocyte populations with clodronate liposomes (Supplementary Fig. S1), we wanted to examine the potential that virus infection could influence the survival of these immune-compromised mosquitoes. While *Ae. aegypti* display increased susceptibility to bacterial challenge following phagocyte depletion[40], clodronate treatment does not impact mosquito survival following blood-feeding or virus infection (DENV/ZIKV) (Supplementary Fig. S3). This suggests that the loss of a subset of granulocytes following clodronate treatment does not have a significant impact on mosquito fitness.

### Virus primarily infects phagocytic granulocyte populations of mosquito hemocytes

Previous studies demonstrate that arboviruses are able to infect mosquito hemocytes[23–27,45], with both prohemocytes[27] and granulocytes[25,26] implicated as the predominant hemocyte subtype infected by virus. To further investigate this immune cell specificity for virus infection, DENV-infected mosquitoes (10 dpi) were injected with fluorescent beads to enable the identification of phagocytic granulocyte populations as previously described[39,41,42,46]. Following perfusion, hemocytes were examined by IFA for the presence/absence of DENV and the uptake of the fluorescent beads. Similar to previous studies[23,26,27], DENV readily infected mosquito hemocytes (Fig. 2a). While approximately 80% of observed hemocytes were positive for DENV, ~70% of hemocytes were positive for both DENV and fluorescent beads which are indicative of phagocytic granulocyte populations (Fig. 2b). Without other reliable cell markers for *Ae. aegypti* hemocytes it is unclear if the remaining DENV positive cells that did not contain fluorescent beads (~6% of cells) are prohemocytes, oenocytoids, or granulocytes that failed to uptake a fluorescent bead. These results suggest that the majority of virus-infected hemocytes are phagocytic. However, due to technical limitations in our ability to define mosquito hemocyte subtypes, our data can only infer that the predominant immune cells involved in virus infection are phagocytic granulocytes, and cannot fully exclude the potential roles of other immune cell populations in virus infection.

With these experiments performed at only a single timepoint, we further examined hemocyte virus infection temporally under biological conditions to determine the percentage of virus-infected hemocytes during the course of DENV infection. At early stages of infection (Day 2 and 4), when virus remains localized to the midgut, there was no infection in circulating hemocytes (Fig. 2c). However, beginning at 6 dpi, hemocytes were found infected with DENV, and increased in prevalence at 8 and 10 dpi (Fig. 2c). This suggests that hemocyte infection corresponds with the approximate timing of virus dissemination from the midgut[23,47,48].

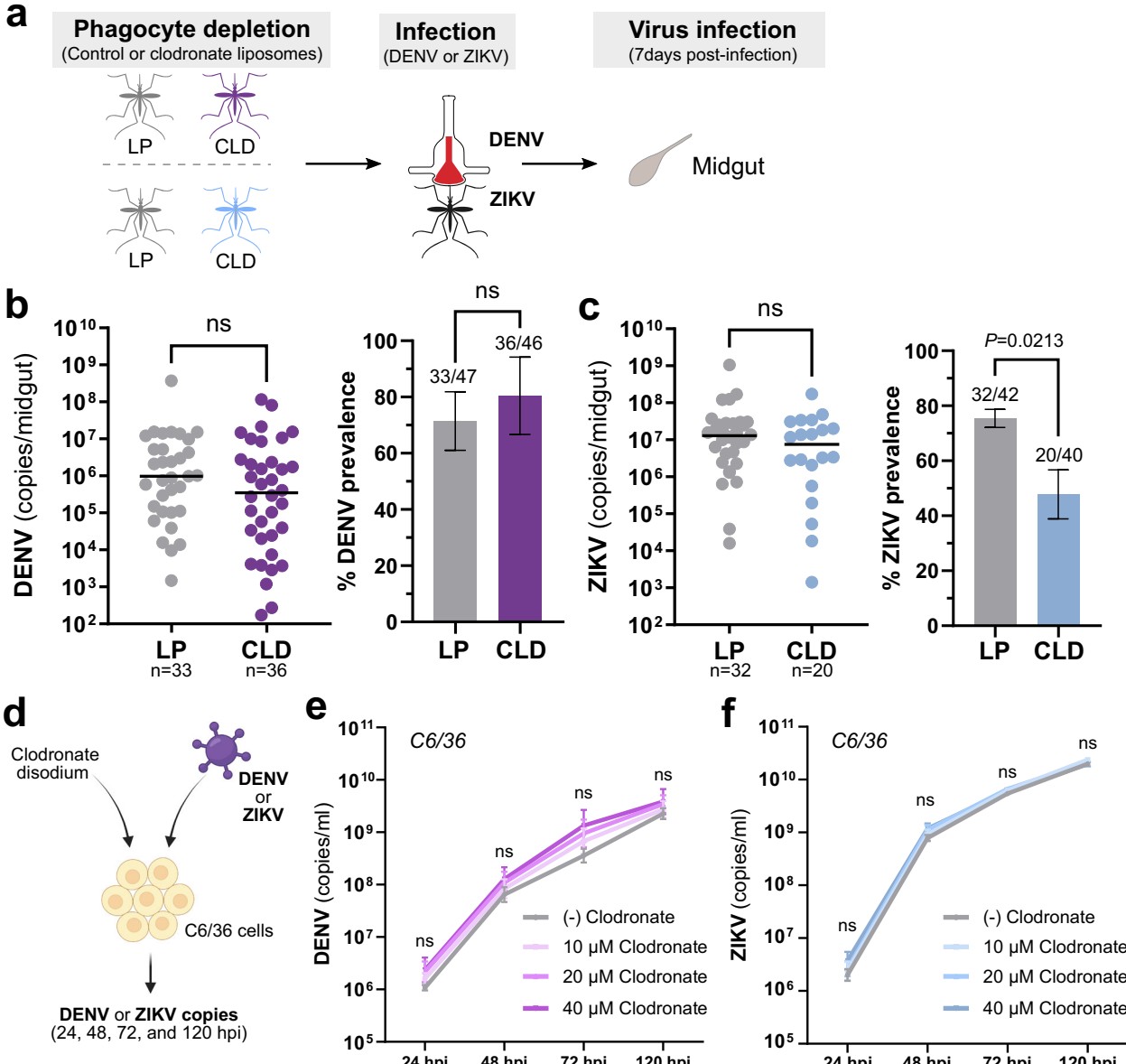

**Fig. 1 | Effects of phagocyte depletion on DENV and ZIKV midgut infection.** Overview of experiments performed in *Ae. aegypti* where adult female mosquitoes were injected with either control (LP) or clodronate liposomes (CLD) to examine the effects of phagocytic granulocyte depletion on midgut virus infection (**a**). Approximately 24 h post-injection, mosquitoes were orally infected with DENV or ZIKV, then midguts were dissected at 7 days post-infection to determine infection outcomes. Viral midgut copy numbers and infection prevalence were determined for DENV (**b**) or ZIKV (**c**) by qRT-PCR, with each dot representing the viral titer of each individual midgut, with the median marked by the black line. The infection prevalence (number of infected mosquitoes of the total analyzed) is displayed in bar graphs as the mean ± SEM. All infection data were pooled from three or more independent experiments. Viral copy numbers were analyzed using a two-tailed Mann–Whitney test, while prevalence data were examined using a two-sided

Fisher's exact test. Exact *P* values are displayed in the figure where applicable; ns, not significant. Overview of in vitro experiments performed with C6/36 cells to examine DENV or ZIKV infection in the presence of different concentrations of clodronate disodium salt (**d**), the active component used in clodronate liposomes. DENV (**e**) and ZIKV (**f**) copy numbers were examined at 24, 48, 72, 120 h post-infection (hpi). Virus growth curves were examined using a two-way ANOVA with a Dunnett's multiple comparison to determine significance. Data in (**e**) and (**f**) display the mean ± SEM of a total of six replicates obtained from two independent biological experiments. Statistical analysis was performed using a two-way ANOVA with a Dunnett's multiple comparisons test. ns not significant. Illustrations in (**a**) were created by David Hall using Inkscape, while the image in (**d**) was created in BioRender. Smith, R. (2025) https://biorender.com/miahijk. Source data are provided as a Source Data file.

## Phagocytic granulocytes adhere to tissues involved in arbovirus transmission

Given the large proportion of circulating hemocytes infected by virus (Fig. 2) and the propensity for hemocytes to adhere to multiple mosquito tissues[22,45], the adherence of virus-infected hemocytes to other mosquito tissues may promote virus infection. Arguably, the two most important tissues for mosquito arbovirus transmission are the salivary glands, which must become infected for transmission via saliva to a

vertebrate host during a blood meal, and the ovaries, which are essential for vertical transmission of virus to offspring. However, the ability of hemocytes to attach to either the salivary glands or ovaries has not been previously characterized. For this reason, we injected mosquitoes with CM-DiI to selectively label hemocytes[46,49] and fluorescent beads to further identify phagocytic granulocytes as in earlier experiments (Fig. 2). We then examined dissected ovaries and salivary glands from mosquitoes at 7 dpi for the presence of attached

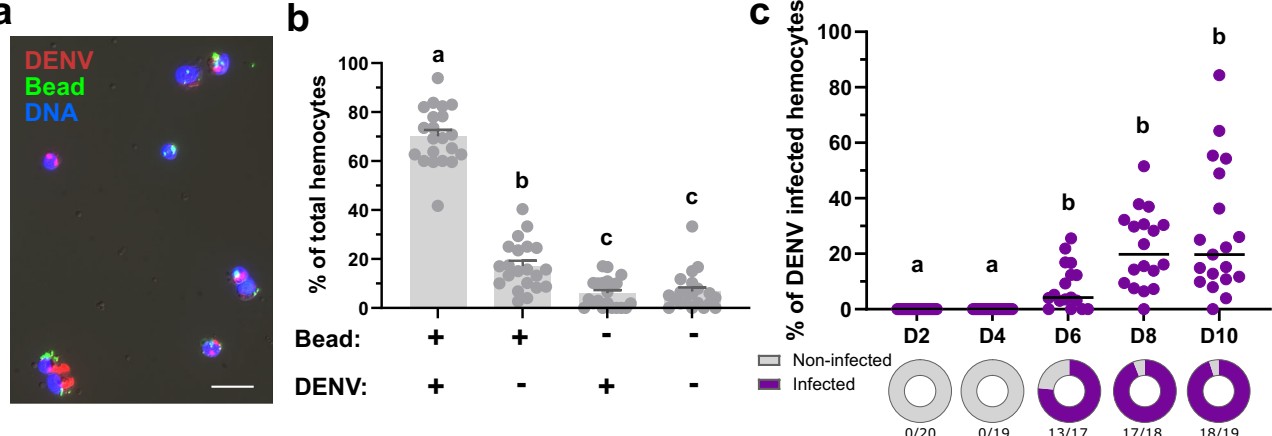

**Fig. 2 | Immunolocalization of DENV in phagocytic granulocytes.** Hemocytes were perfused from DENV-infected mosquitoes 10 days post-infecton after injection with fluorescent beads (green). Following fixation, virus localization was examined by immunofluorescence on fixed hemocytes using an anti-DENV monoclonal antibody (clone 3H5-1) followed by an Alexa Fluor 568 goat anti-mouse IgG secondary antibody (red; scale bar: 10 µm) and mounted in ProLong®Diamond Antifade mountant with DAPI (blue) (**a**). To examine the abundance of virus in phagocytic granulocyte populations, the presence/absence of DENV was quantified in the context of the presence/absence of beads used to identify phagocytic granulocyte populations. **b** For each experimental outcome, the percentage of the total number of hemocytes for each phenotype are displayed per individual mosquito (dots, *n* = 21) and displayed as the mean ± SEM. Data were pooled from two independent biological experiments. **c** Additional experiments examine the percentage of virus-infected hemocytes following an oral infection with DENV. Hemocytes were perfused at 2, 4, 6, 8, and 10 days (D) post-infection, fixed, then examined by immunofluorescence as outlined above. Circles under each timepoint display the prevalence of infection and the number of individual mosquitoes (*n*) examined from two independent biological experiments. For both (**b**) and (**c**), statistical analysis was performed using Kruskal–Wallis with a Dunn's multiple comparison test. Letters denote statistically significant differences between sample treatments. Source data and more detailed statistical comparisons, including *P* values associated with each comparison, are provided as a Source Data file.

phagocytic granulocytes to these tissues using fluorescent microscopy. In both DENV- (Fig. 3a) and ZIKV-infected mosquitoes (Supplementary Fig. S4), we observed phagocytic hemocytes attached to the ovaries and salivary glands, suggesting that adherent granulocytes may facilitate the transport of virus to these secondary tissues required for transmission. However, one caveat of these experiments is that bead injection increased hemocyte attachment to relevant mosquito tissues such as the salivary gland, ovary, and midgut (Fig. 3b).

To determine how hemocyte attachment to mosquito tissues was influenced by different physiological conditions in vivo, we examined hemocyte attachment to salivary gland, ovary, and midgut tissues under naïve, blood-fed, and DENV-infected conditions at multiple timepoints (Fig. 3c, Supplementary Fig. S5). Hemocyte attachment to the salivary glands was unaffected by blood-feeding or infection and displayed consistent levels of attachment across the time periods examined (Fig. 3c). In contrast, blood-feeding promoted a significant increase in hemocyte attachment to the ovaries, independent of infection status, which remained elevated compared to naïve conditions up to 10 days post-feeding (Fig. 3c). However, following blood-feeding, the ovaries undergo pronounced changes in size as a result of vitellogenesis, such that it is unclear if these changes simply represent the increased surface area corresponding with egg production. Similar to the salivary glands, hemocyte attachment to the midgut was relatively unaffected by physiological conditions, with significant changes in attachment only found between naive and blood-fed conditions at 10 days post-feeding (Fig. 3c). Together, these data support that hemocyte attachment occurs regularly to salivary gland, ovary, and midgut tissues in the mosquito host.

### Granulocyte depletion attenuates virus infection
Since granulocytes become infected with virus (Fig. 2) and can attach to the ovaries and salivary glands (Fig. 3), we performed experiments to examine the role of hemocytes in virus infection to peripheral mosquito tissues. To bypass any potential influence of phagocyte depletion on midgut infection outcomes, mosquitoes were first

challenged with DENV or ZIKV, then treated with control or clodronate liposomes three days post-infection (Fig. 4a). The influence of phagocyte depletion on virus infection was examined in peripheral tissues by measuring virus copies in legs, ovaries, and salivary glands (Fig. 4a). DENV RNA copies were significantly reduced in the legs of CLD-treated mosquitoes at 8, 10, and 12 dpi, the ovaries at 12 dpi, and the salivary glands at 10 and 12 dpi (Fig. 4b). In addition, there was a significant decrease in the prevalence of DENV infection following granulocyte depletion at day 8 post-infection in the legs, days 8 and 10 post-infection in the ovaries, and day 10 post-infection in the salivary glands (Fig. 4c). Similarly, for ZIKV-infected mosquitoes, CLD treatment significantly reduced ZIKV copies in the legs at 8 dpi, the ovaries at 8 and 10 dpi, and the salivary glands at 8 dpi (Fig. 4d). This corresponds with a significant decrease in the prevalence of infection on both 8 and 10 dpi in the legs, as well as 8 dpi in the ovaries and salivary glands (Fig. 4e). Additional focus forming assays (FFAs) confirm the viability of DENV and ZIKV infection of the ovaries and salivary glands, providing further validation of our qRT-PCR data that support the influence of granulocyte depletion on infection prevalence (Supplementary Fig. S6). Together, these results indicate that phagocyte depletion delays virus dissemination from the midgut to the legs, ovaries, and salivary glands, providing strong evidence that granulocytes enhance DENV and ZIKV infection of peripheral mosquito tissues.

### Phagocytic granulocytes promote virus infection of uninfected tissues
While evidence suggests that virus infects mosquito granulocytes (Fig. 2) and the depletion of these immune cells attenuates virus infection to peripheral tissues (Fig. 4), it remains unclear whether virus-infected granulocytes represent a viable tropism for virus and if they are able to directly confer an infection to other mosquito tissues. Moreover, the relative contributions of virus-infected hemocytes in virus infection compared to free virus circulating in the mosquito hemolymph has not been previously evaluated. To approach these

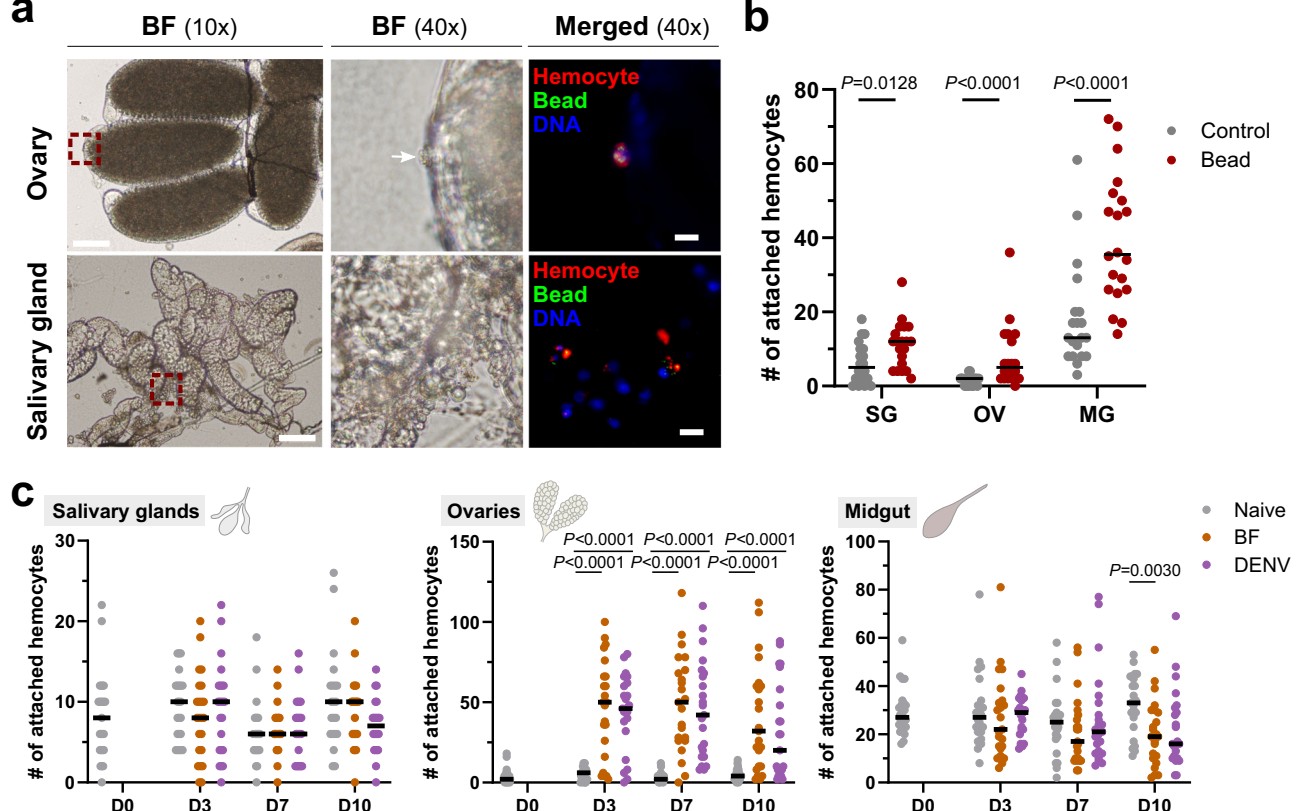

**Fig. 3 | Phagocytic granulocytes associate with the salivary glands and ovaries of DENV-infected mosquitoes.** To examine hemocyte attachment to mosquito tissues, DENV-infected mosquitoes were injected with CM-DiI (red) and fluorescent beads (green) at 7 days post-infection to identify phagocytic granulocyte populations (**a**). Following staining in vivo, ovary and salivary gland tissues were dissected to examine granulocyte attachment to each respective tissue and mounted using ProLong®Diamond Antifade mountant with DAPI (blue). Red dashed line boxes denote the field of view at 40× magnification, with white arrows used to indicate attached phagocytic hemocytes in the bright field (BF) image where applicable. Scale bars denote 100 μm for 10× images and 10 μm for 40× images. **b** The number of hemocytes attached to salivary gland (SG), ovary (OV), and midgut (MG) tissues was quantified in the presence or absence of beads in individual mosquitoes (*n* = 20 for all tissues, two independent replicates). Statistical analysis were performed using Multiple Mann–Whitney tests with a two-stage step-up (Benjamini, Krieger, and Yekutieli) to correct for multiple comparisons. Exact *P* values are displayed for each comparison. Since bead injection artificially increased hemocyte attachment (**b**), similar experiments relying on only CM-DiI staining were used to quantify hemocyte attachment to the salivary glands, ovaries, and midgut under naïve, blood-fed (BF), and DENV infection (**c**). Attachment was evaluated at 0, 3, 7, and 10 days (D) post blood-feeding, infection, or in age-matched naïve controls in individual mosquitoes (*n* = 20 for all tissues, two independent replicates). Statistical analysis were performed using Multiple Mann–Whitney tests with a two-stage step-up (Benjamini, Krieger, and Yekutieli) to correct for multiple comparisons. Exact *P* values are displayed for any significant comparisons. For both (**b**) and (**c**), dots represent data collected from individual mosquitoes. Tissue images in (**c**) were created by David Hall using Inkscape. Source data are provided as a Source Data file.

questions, we performed experiments transferring either acellular or cellular fractions of perfused hemolymph from virus-infected mosquitoes to naïve mosquitoes (Fig. 5a). Hemolymph was perfused from DENV- or ZIKV-infected mosquitoes at 10 days post-infection, then separated by centrifugation into supernatant (SUP) and cellular fractions (CELL) prior to injection into naïve mosquitoes similar to previously published studies[50,51] (Fig. 5a). When whole mosquitoes were evaluated at 1, 2, or 4 days post-transfer for DENV or at 4 days post-transfer for ZIKV, mosquitoes receiving the cellular fraction had a significantly higher prevalence of DENV or ZIKV at 4 days post-transfer (Fig. 5b) when compared to mosquitoes receiving the acellular hemolymph supernatant fraction. Moreover, the transfer of the cellular fraction resulted in the infection of DENV or ZIKV in the salivary glands, ovary, midgut, and carcass tissues (Fig. 5c). While these results were collected by qPCR, experiments in which DENV- or ZIKV-infected tissue samples were examined using FFAs confirm the viability of these infections (Supplementary Fig. S7).

Based on the observation that few mosquitoes injected with the acellular hemolymph supernatant developed an infection (~5%) (Fig. 5b), we further examined virus copy numbers and the ability of

both fractions to infect mosquito cells (Fig. 5d). No significant differences in viral genome copy numbers were detected between acellular hemolymph and cell fractions in DENV or ZIKV experiments (Fig. 5e). To determine if there were differences in the infectiousness of virus isolates obtained from the supernatant or cell fractions, both isolates were added to cultures of C6/36 cells, with infection outcomes monitored at 2- and 3-days post-infection (Fig. 5f). While DENV cell fractions developed a productive infection, virus collected from the supernatant fraction resulted in a lower prevalence of infection (Fig. 5f). In contrast, both the supernatant and cell fractions for ZIKV displayed similar infection outcomes (Fig. 5f), despite the cell fraction resulting in a higher prevalence of infection in mosquitoes during transfer experiments (Fig. 5b). While we cannot exclude the potential that the collection of the acellular supernatant fraction (perfusion, centrifugation, etc.) may have influenced DENV infectivity, the observed differences between in vitro and in vivo infectivity of the ZIKV acellular and cellular fractions suggest that hemocytes may provide a more efficient method to promote virus infection.

With our previous results suggesting that phagocytic granulocytes are an important factor in virus infection (Figs. 2–5), we repeated

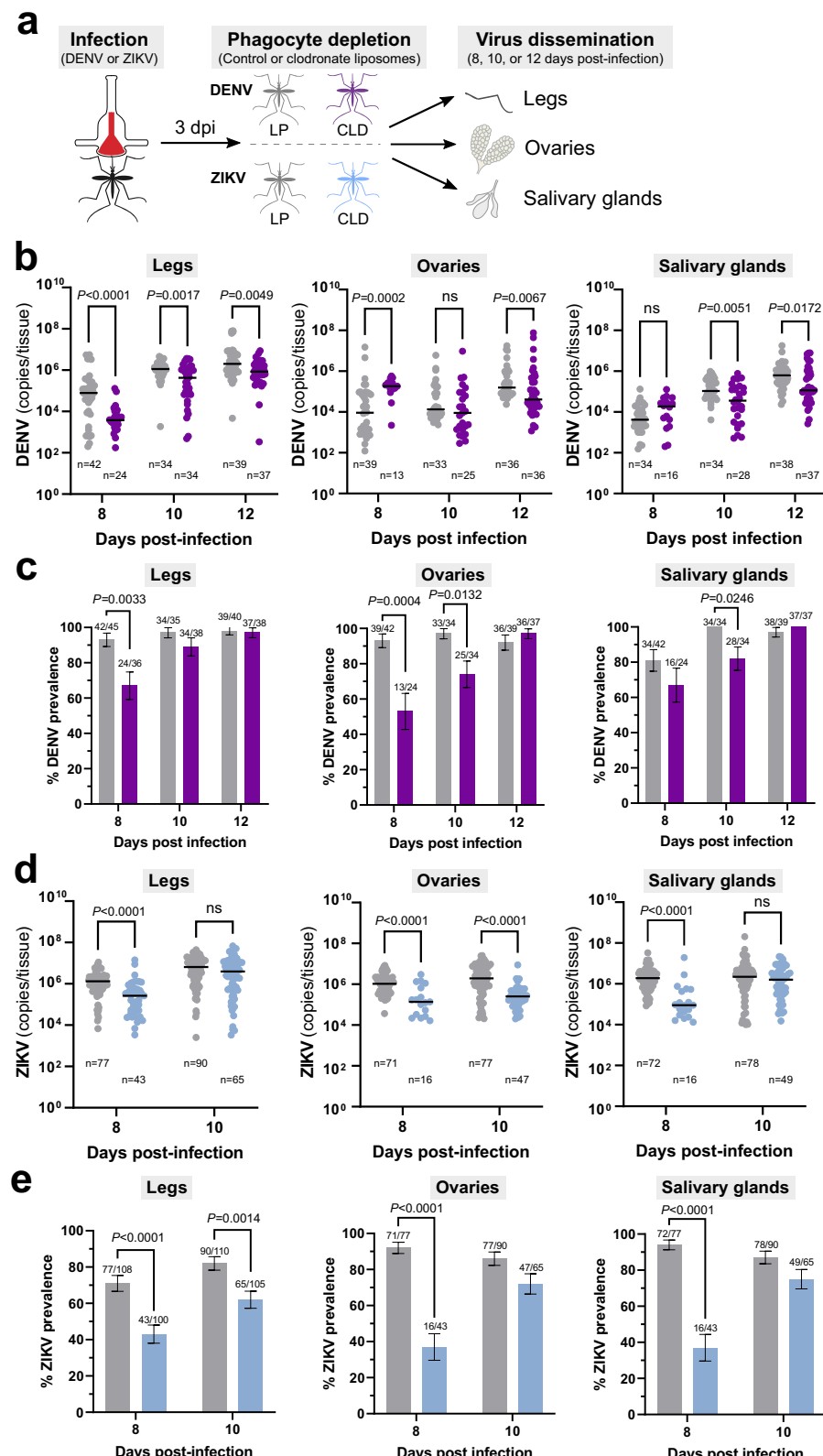

our transfer experiments in the context of phagocyte depletion to determine if granulocytes are the cell type responsible for the successful viral infection of the naïve mosquito host following transfer (Fig. 6a). Similar to our earlier transfer experiments (Fig. 5b), transfer of the cell fraction from control mosquitoes injected with LP produced a virus infection in naïve mosquitoes when examined 4 days post-transfer for either DENV or ZIKV (Fig. 6b). In contrast, when samples

originated from CLD-treated mosquitoes, the transfer of virus was completely abolished for both DENV and ZIKV (Fig. 6b). Similar to previous results (Fig. 5b), the supernatant fraction was unable to efficiently transfer a viral infection to naïve mosquitoes (Fig. 6b). Taken together, these results provide strong evidence that granulocytes are capable of acquiring and promoting the infection of flaviviruses to mosquito tissues required for arbovirus transmission.

**Fig. 4 | Phagocytic granulocyte depletion impairs virus infection of the mosquito legs, ovaries, and salivary glands. a** Overview of virus dissemination experiments. After oral challenge with DENV or ZIKV, mosquitoes were injected with control (LP) or clodronate liposomes (CLD) at 3 days post-infection. Virus dissemination was examined in legs, ovaries, and salivary glands at 8, 10, and 12 days post-infection for DENV (**b, c**), or 8- and 10-days post-infection for ZIKV (**d, e**). Virus copy numbers from infected samples are shown for the legs, ovaries, and salivary glands for each time point for DENV (**b**) or ZIKV (**d**), with each dot representing the viral titer of each individual mosquito sample (*n*) and the median marked by the black line. The infection prevalence (number of infected tissues of the total analyzed) is displayed for the legs, ovaries, and salivary glands in bar graphs as the mean ± SEM for DENV (**c**) or ZIKV (**e**). Infection data were pooled from four independent experiments for DENV (**b**) and two independent experiments for ZIKV (**d**). Viral copy numbers were analyzed using a Multiple Mann–Whitney test with a two-stage step-up (Benjamini, Krieger, and Yekutieli) to correct for multiple comparisons. Exact *P* values are displayed in the figure where applicable; ns, not significant. Infection prevalence data were analyzed using a two-sided Fisher's exact test. Exact *P* values are displayed in the figure for any significant comparisons. Illustrations in (**a**) were created by David Hall using Inkscape. Source data are provided as a Source Data file.

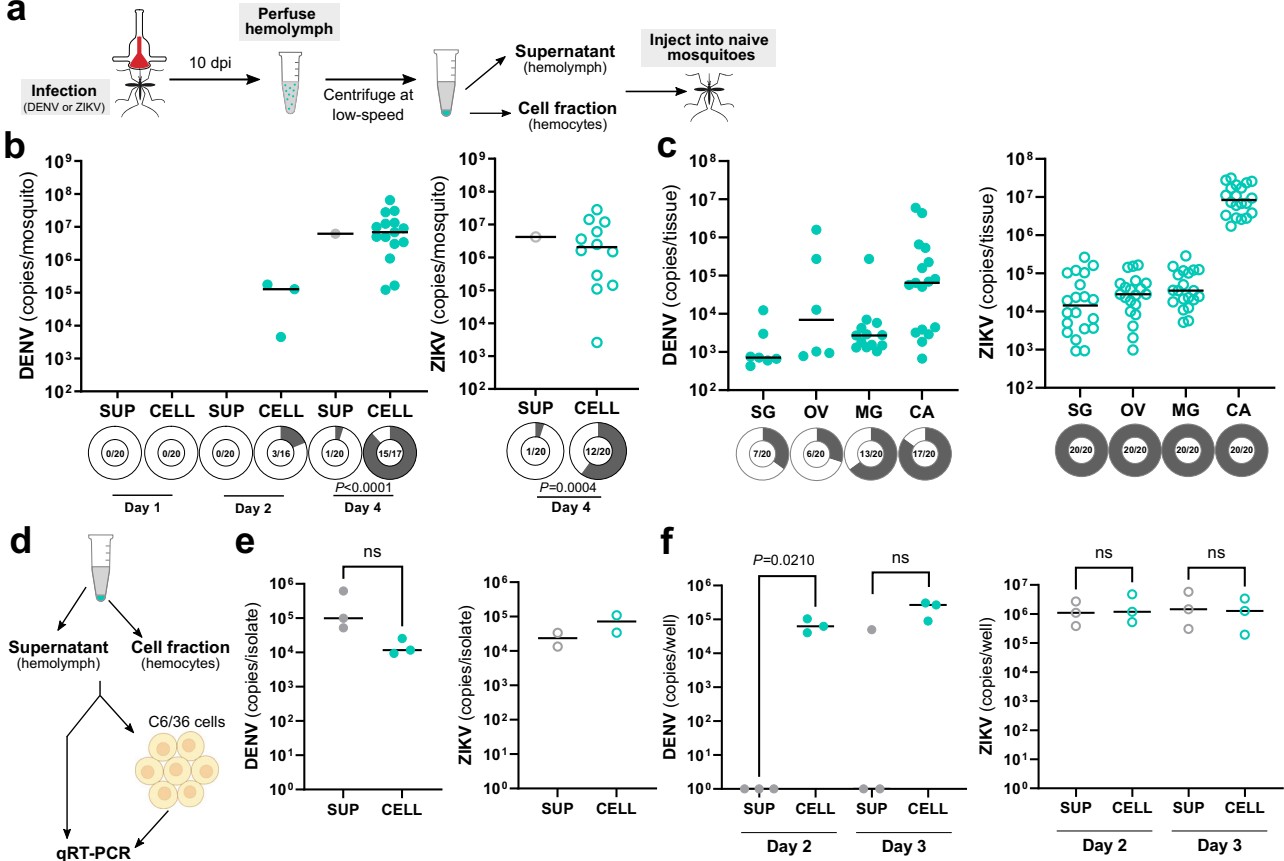

**Fig. 5 | Cellular fractions of the hemolymph are more efficient in transferring a virus infection to naïve mosquitoes.** To assess the respective infectivity of the cellular and acellular fractions of mosquito hemolymph, *Ae. aegypti* were first infected with DENV or ZIKV, then hemolymph was perfused from mosquitoes at 10 days post-infection (**a**). Hemolymph was separated into hemocyte-containing cell (CELL) and acellular hemolymph supernatant (SUP) fractions using low-speed centrifugation and then injected into naïve mosquitoes (**a**). Whole mosquitoes were evaluated for viral titer and infection prevalence by qRT-PCR at 1, 2, and 4 days post-injection for DENV and 4 days post-injection for ZIKV (**b**). Additional analysis of salivary gland (SG), ovary (OV), midgut (MG), and carcass (CA) tissues at 4 days post-transfer of the CELL fraction confirm DENV and ZIKV infection of each respective tissue (**c**). For both (**b**) and (**c**), dots represent the viral titer of each individual mosquito/tissue, with the black line denoting the median. Due to the low numbers of virus positive samples in certain control and experimental conditions, viral titers were not directly compared. Infection prevalence (number of infected mosquitoes of the total examined, *n*) is displayed as shaded areas of circles under each experimental condition. Prevalence data were analyzed using a two-sided Fisher's exact test, with exact *P* values displayed in the figure where applicable. Data were pooled from two independent experiments. **d** Overview of experiments to examine the SUP and CELL fractions, where the DENV and ZIKV viral copy numbers were compared in SUP and CELL fractions (**e**) or that examine the ability of the SUP and CELL fractions to infect C6/36 cells at Days 2 and 3 post-infection (**f**). ZIKV viral copy numbers in **e** represent two independent experiments. All other data in (**e**) and (**f**) represents three independent biological experiments, and were analyzed using a two-tailed Student's *t* test. Exact *P* values are displayed in the figure where applicable; ns, not significant. Illustrations in **a** were created by David Hall using Inkscape. The image in (**d**) was created in part using BioRender. Smith, R. (2025) https://BioRender.com/9cvyzdr and illustrations created by David Hall using Inkscape. Source data are provided as a Source Data file.

## Discussion

While immune cells are instrumental to DENV and ZIKV infection and dissemination in vertebrate systems[32–38], we still lack a critical understanding of the mechanisms that drive virus infection and dissemination in the mosquito host. These efforts have previously been hindered by the lack of genetic tools to manipulate mosquito hemocyte populations, thereby limiting our ability to study these immune cells and their contributions to arbovirus infection. By taking advantage of recently described methods to deplete phagocytic hemocyte populations in arthropods[39–41], we demonstrate that phagocytic hemocytes

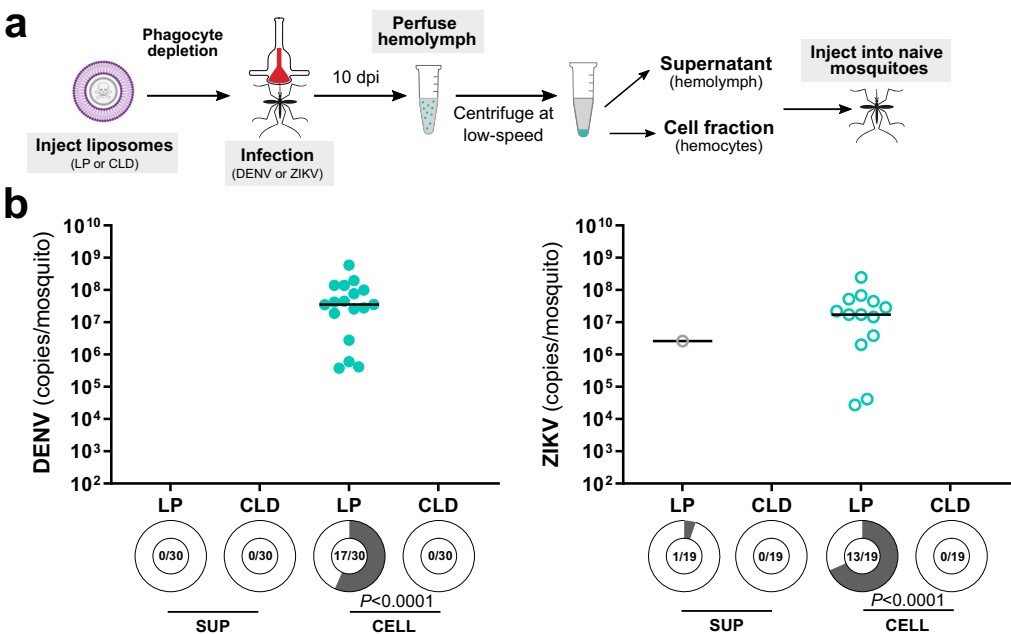

**Fig. 6 | Phagocytic granulocyte depletion impairs the transfer of a virus infection to naive mosquitoes.** Mosquitoes were treated with clodronate liposomes (CLD) to deplete phagocytic granulocytes prior to DENV or ZIKV infection to confirm that granulocytes are required for the transfer of virus infection in the cellular fractions of the mosquito hemolymph (**a**). Empty liposomes (LP) serve as a control. Whole mosquitoes were assessed for viral titer and infection prevalence at 4 days post-injection for both DENV and ZIKV (**b**). Due to the low numbers of virus positive samples in certain control and experimental conditions, viral titers were not directly compared. Dots represent the viral copy number of individual mosquito samples, with the median displayed by the black line. Shaded areas of circles under each experimental condition correspond to infection prevalence, with the number infected of the total number of mosquitoes examined (n) displayed for each treatment. Prevalence data were analyzed using a two-sided Fisher's exact test, with exact $P$ values displayed in the figure where applicable. Data were pooled from two independent experiments. The image in **a** was created in part using BioRender. Smith, R. (2025) https://BioRender.com/4p4wl03 and illustrations created by David Hall using Inkscape. Source data are provided as a Source Data file.

support arbovirus infection and facilitate further virus infection to important tissues such as the salivary glands and ovaries in the mosquito *Ae. aegypti*.

Our phagocyte depletion experiments suggest that DENV and ZIKV titers in the mosquito midgut are not directly influenced by the presence or absence of phagocytic granulocyte populations. However, the reduced prevalence of midgut ZIKV infection following granulocyte depletion suggests that these immune cells may still contribute to the success of midgut infection, potentially through the regulation of epithelial homeostasis[52,53]. Similar pro-viral effects of phagocytic granulocytes on midgut virus infection have been previously described, yet in contrast to our own study, previous studies show reduced DENV and ZIKV midgut titers when populations of phagocytic granulocytes are functionally overloaded via bead injection[26]. While both methodologies target granulocytes, bead injection may also promote other unintended cellular or humoral responses that reduce midgut infection or virus replication, as evidenced by an increase in midgut-associated hemocytes after bead injection[26]. While further study is required to fully delineate the complex role of hemocytes in midgut viral infection and replication, our results suggest that the overall influence of granulocytes on midgut infection are minimal and perhaps virus- or titer-dependent.

Similar to previous studies[12,23–26], we observed virus localization in circulating hemocytes collected from perfused hemolymph. However, there have been disagreements as to the immune cell subtypes infected by virus, with previous studies implicating either prohemocytes[27] or granulocytes[25,26]. Through immunofluorescence experiments pairing virus infection with the uptake of fluorescent beads, we

demonstrate that the majority of virus-infected hemocytes are phagocytic granulocytes. However, one limitation of these immunofluorescence assays that involve fixation is that other hemocyte subtypes (prohemocytes and oenocytoids) are not adherent and represent a small proportion of hemocytes following fixation[39]. Therefore, without additional hemocyte subtype-specific markers to denote these cell types, the potential remains that other hemocyte subtypes may also be infected by virus as previously suggested[27]. Yet, when paired with our clodronate liposome experiments that demonstrate roles of phagocytic granulocytes in virus dissemination and adoptive transfer experiments, we believe that granulocytes are a focal point for virus infection in the hemolymph. With the existence of multiple granulocyte subtypes[39,42,54,55], additional experiments are required to determine if specific subpopulations of granulocytes differ in their susceptibility to virus infection.

While our work and that of others has demonstrated that virus can infect the hemocytes of mosquitoes and other insects[12,23–26,45], the exact mechanism and location of hemocyte virus acquisition remains unknown. Previous work has shown that hemocytes adhere to multiple mosquito tissues[45,49] and are recruited to the midgut following virus infection[26]. In our experiments examining hemocyte attachment at various timepoints and physiological conditions, we observe relatively high levels of attachment under naïve conditions, with data suggesting that hemocyte attachment to the midgut is not influenced by blood-feeding or virus infection. However, the abundance of non-specific hemocyte attachment may enable hemocytes associated with the midgut to potentially acquire an infection after direct contact with the midgut basal lamina, where mature virions are concentrated[15]. In

support of this hypothesis, our data suggest that hemocytes become infected with DENV at ~6 dpi, the approximate timing of *Flavivirus* dissemination from the midgut[23,47,48]. With hemocyte attachment to the midgut and other tissues believed to be transient[56,57], an infected sessile hemocyte may transition to enter circulation and potentially spread an infection to other mosquito tissues. Other studies have demonstrated that arboviruses infect the tracheal network associated with the midgut epithelium[23,58,59], a tissue (trachea) which mosquito hemocytes associate[45], potentially providing other routes of infection similar to that required for hemocyte infection and baculovirus dissemination in Lepidoptera[12,60]. Alternatively, hemocytes may also acquire infection through the uptake of free virus in the hemolymph after escape from virus-infected tissues. Given these multiple possible routes of infection, further investigation is required to determine the mechanisms by which mosquito hemocytes become infected and further spread a virus infection in the mosquito host.

When clodronate depletion experiments were performed after virus midgut infection to bypass any effect on midgut infection outcomes, we found that phagocytic granulocyte populations have an integral role in the spread of virus infection. Phagocyte depletion reduced infection prevalence and viral copy numbers in the legs, a tissue routinely used as an indicator of transmission potential[61]. In addition, phagocyte depletion lowered the prevalence and copy number of DENV and ZIKV in the ovaries and salivary glands, the infection of which can respectively lead to vertical transmission to offspring or transmission to a vertebrate host during blood-feeding. For DENV, observations of reduced infection prevalence in secondary tissues was most prominent at earlier time points (8 or 10 dpi), with the reduction of DENV copies the largest at 12 dpi for both ovary and salivary gland tissues. We observed similar patterns for ZIKV infection, where infection prevalence and viral copy numbers in secondary tissues were most impacted at 8 dpi across tissues. While this suggests that there may be some slight differences in the kinetics of virus dissemination between DENV and ZIKV, our data suggest that phagocytic granulocytes have an important, although not absolute, role in the spread of virus infection in the mosquito host.

These data indicate that phagocytic granulocytes attach to both the ovaries and salivary glands in DENV- and ZIKV-infected mosquitoes and that phagocytic granulocyte populations have significant contributions to DENV and ZIKV infection in the mosquito, likely enhancing virus transmission to both the vertebrate host and invertebrate offspring. Despite this seemingly pro-viral role for phagocytic granulocytes, it should be noted that the loss of these immune cell populations only attenuates the spread of virus infection and does not eliminate the dissemination of virus to other mosquito tissues. At present, it is unclear if this is due to incomplete phagocyte depletion using clodronate liposomes, the presence of other infected hemocyte subtypes, free virus in the hemolymph, or if additional, yet unexplored, mechanisms of virus dissemination occur in the mosquito host.

Through the use of transfer experiments using either acellular or cellular hemolymph fractions, we demonstrate that virus-infected hemocytes are capable of conferring a productive viral infection to a naïve host. Through clodronate depletion, we specifically implicate phagocytic granulocytes in this virus transfer, providing strong support for the importance of these immune cells in mediating virus infection of mosquito tissues. In contrast, we found that the acellular hemolymph was far less efficient in transferring a productive infection to naïve mosquitoes despite containing comparable virus loads when measured by qRT-PCR. While we cannot exclude that the perfusion and centrifugation methods used to isolate the acellular hemolymph may have altered virus infectivity, the supernatant fractions for both DENV and ZIKV were still able to occasionally infect in vivo following transfer experiments and were able to promote infection in C6/36 cells in vitro with varying levels of success depending on the specific virus. Since ZIKV was able to infect C6/36 cells more efficiently than DENV, it is

possible that a significant proportion of DENV collected from the hemolymph may be immature virus[62–64], which may contribute to the low levels of infection observed for DENV both in vitro and in vivo. However, there may be additional unknown factors present in the mosquito hemolymph that may also restrict the infectivity of free virus. While these questions are of significant interest for future study, these data suggest that hemocyte virus infection enhances the success of virus dissemination.

As a result, virus-infected granulocytes may behave as a "trojan horse" to protect intracellular virus from antiviral factors present in the hemolymph. With evidence that hemocytes can adhere to various tissues and again re-enter circulation[56,57], an infected hemocyte may inadvertently deliver virus to target tissues (such as the salivary glands and ovaries) through circulation in the hemolymph. Furthermore, hemocytes may also serve as a cellular tropism for further virus replication, as previously suggested[18,25]. Therefore, virus-infected hemocytes may provide the cellular machinery needed to further amplify virus as a secondary site of replication after midgut infection to further enhance the spread of virus to other mosquito tissues. However, additional studies are required to definitively confirm virus replication in hemocytes and to better understand how hemocytes may directly or indirectly contribute to the dissemination of virus infection.

Similar roles in arbovirus infection have also been described for vertebrate immune cell populations. While tissue-resident dendritic cells (such as Langerhans cells) provide sites of initial arbovirus replication and local dissemination[65–68], monocytes and monocyte-derived macrophages transport virus through the circulatory system to further enhance virus replication, persistence, and dissemination[38,66,69]. This includes comparable function of immune cells as "trojan horses" in vertebrate systems, where monocyte/macrophage populations mediate viral entry and dissemination in the central nervous system[69]. With monocytes/macrophages performing as professional phagocytes similar to that of mosquito granulocytes, we believe that this represents an exciting parallel between vertebrate and invertebrate systems in the manner by which virus exploit immune cell populations for their own replication and propagation.

In addition to the roles of mosquito hemocytes in promoting virus infection outlined in this study, hemocytes have also been implicated in anti-viral immunity. Previous studies in *Ae. aegypti* demonstrate that impairing hemocyte function by overloading these cells with beads increased viral titers in whole mosquitoes injected with ZIKV or DENV[26], implying that phagocytic activity in hemocytes restricts virus dissemination. Similar observations have also been described for *Drosophila*, where the uptake of virus RNA by hemocytes mediates a systemic RNAi anti-viral response limiting virus replication[31]. Therefore, it is likely that hemocytes play a complex role in mosquito virus infection and may be involved in both anti-viral and pro-viral activities that may vary among specialized hemocyte sub-populations.

While our study examines the influence of hemocytes in flavivirus (DENV and ZIKV) dissemination, the role of hemocytes in virus infection and dissemination may differ for other arboviruses. For instance, alphaviruses and flaviviruses have different strategies for replication and assembly[70–72] where alphaviruses typically replicate faster in the mosquito host. For example, chikungunya virus is able to quickly replicate and escape the midgut as early as 24 h after an infectious blood meal[15,73], contrasting the approximate three days required for the earliest escape of ZIKV from the midgut[16]. As a result, alphaviruses may be able to exploit transient damage to the basal lamina resulting from blood-feeding prior to its repair, potentially enabling free virus to escape into the hemolymph and disseminate to other mosquito tissues without the need of hemocytes. However, another alphavirus, Sindbis virus, has been shown to infect and replicate within hemocytes[25], implying that hemocytes may have similar roles in alphavirus dissemination. Further investigation is required to more precisely

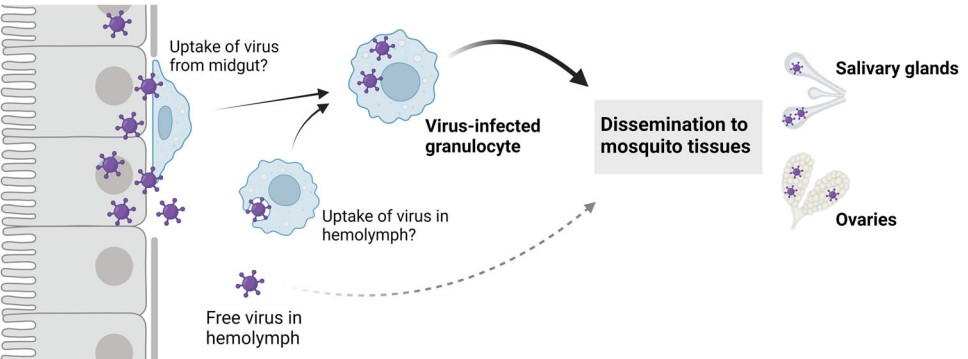

**Fig. 7 | "Trojan horse" model of virus infection in the mosquito host.** Following virus infection of the mosquito midgut, a subset of phagocytic granulocytes likely acquire virus either through attachment to the virus-infected midgut, the uptake of free virus in the mosquito hemolymph, or the attachment to other infected mosquito tissues such as the trachea (not shown). Through the movement of these immune cells in the hemolymph and their ability to adhere to mosquito tissues, virus-infected granulocytes display increased efficiency to promote virus infection of the salivary glands and ovaries when compared to free virus present in the hemolymph. This suggests that these virus-infected granulocytes act as a "trojan-horse" to potentially enable further virus replication and enhance virus infection. Created in BioRender. Smith, R. (2025) https://BioRender.com/kj6zzk3.

determine whether the roles of hemocytes in arbovirus infection are conserved for all mosquito-borne arboviruses.

In summary, the results of this study support three main conclusions. First, phagocytic granulocytes are focal points for virus infection and attach to mosquito tissues involved in arbovirus transmission. Second, the depletion of granulocytes via clodronate liposomes attenuates viral infection, implying that granulocytes significantly contribute to the dissemination of virus to tissues such as the salivary glands and ovaries. Third, virus-infected granulocytes are able to transfer a viral infection to a naïve mosquito more efficiently that cell-free hemolymph. Taken together, the results of this study converge upon a model for virus dissemination whereby hemocytes acquire virus and subsequently transport an infection via the hemolymph to other mosquito tissues (Fig. 7). While the exact mechanisms through which hemocytes acquire and promote virus infection remain to be explored, the hemocyte tropism for viruses described here suggest that hemocytes have profound impacts on mosquito vector competence and the extrinsic incubation period for both DENV and ZIKV. The results of this study significantly advance our understanding of virus infection dynamics in mosquitoes that highlight conserved roles of immune cells in virus infection.

## Methods

### Cell culture and virus isolates
*Ae. albopictus* C6/36 cell maintenance and virus culture were performed at Iowa State University (ISU) and the Connecticut Agricultural Experiment Station (CAES) with slight modifications. At ISU, *Ae. albopictus* C6/36 cells were maintained in 25 cm² flasks at 28 °C in L-15 media supplemented with 10% fetal bovine serum (FBS), 4 mM L-glutamine, 100 units/ml penicillin, and 100 μg/ml streptomycin. At CAES, *Ae. albopictus* C6/36 cells were grown in 75 cm² flasks using minimum essential medium containing 2% FBS, 1X non-essential amino acids, 100 U/ml penicillin, and 100 μg/ml streptomycin. Virus infection experiments at both locations were performed with DENV serotype 2 (125270/VENE93; GenBank Accession No. U91870) and ZIKV (PRVABC59; GenBank Accession No. KU501215). To infect cells with virus, growth media was removed and replaced with 2 ml L-15 media supplemented with 2% fetal bovine serum, 4 mM L-glutamine, and 250 μl frozen DENV or ZIKV stock, and left at room temperature rocking periodically for 15 min. The flask was then placed at 28 °C for 45 min rocking periodically, after which 3 ml of additional media supplemented with 2% FBS and 4 mM L-glutamine was added. The infection was allowed to proceed until significant cytopathic effects

were observed, ~4 days for DENV and 5 days for ZIKV. At CAES, 250 μl DENV viral stocks or 250 μl ZIKV were diluted in 3 ml media and used to infect flasks of C6/36 cells at 70–80% confluency. Flasks were placed on a rocking platform with infection media for 1 h at room temperature before adding 12 ml regular media. Infected cells were incubated at 28 °C for 5 days.

### Mosquito rearing and virus infection
All experiments were performed using *Ae. aegypti* (Orlando strain) with slight modifications between performance site locations. At ISU, mosquitoes were reared at 27 °C and 80% humidity with a 16:8 light: dark cycle. Larvae were reared on a diet of ground fish flakes (Tetramin, Tetra), while adult mosquitoes were maintained on a 10% sucrose solution. Mosquito colonies were maintained via artificial membrane feeding using commercial defibrinated sheep blood (HemoStat Laboratories) for egg production. At CAES, mosquitoes were reared at 27 °C and 80% humidity with a 14:10 light: dark cycle. Larvae were reared with a 3:2 mix of powdered liver:yeast, while adults were maintained on 10% sucrose. For virus infection experiments at both locations, adult female mosquitoes (3–5 days after adult emergence) were starved overnight and challenged using a 1:1 mixture of defibrinated sheep blood and fresh DENV or ZIKV culture using a Hemotek artificial membrane feeding system. Virus titers for infection experiments are displayed in Supplementary Table S1. After feeding, mosquitoes were immediately cold anesthetized to allow for the selection of blood-fed mosquitoes.

### Midgut infection experiments
To assess the impact of granulocyte depletion on midgut infection outcomes, naïve mosquitoes (3–5 days post-eclosion) were injected with either 69 nl of control liposomes (LP) or clodronate liposomes (CLD) (Encapsula NanoSciences) at a 1:4 dilution as previously[40]. At 24 h post-injection, mosquitoes were then orally infected with DENV or ZIKV. Midguts were isolated in 1X PBS at 7 days post-DENV or -ZIKV infection, individually homogenized in 400 μl TRIzol Reagent (Invitrogen), and then left at 4 °C overnight to maximize RNA yield. RNA isolation was completed using the Direct-zol RNA MiniPrep kit (Zymo Research) following the manufacturer's protocols. Viral RNA titers were determined by qRT-PCR using the QuantStudio 3 (ThermoFisher) and TaqMan Fast Virus 1-Step Master Mix (ThermoFisher). Reactions were set up according to the manufacturer's protocols using a 10 μl reaction volume and 25 ng of total RNA template. qRT-PCR reactions were performed using previously described primers for DENV[74] or

ZIKV[75] (Supplementary Table S2) with the following PCR conditions: 50 °C for 5 min, 95 °C for 20 s, and 40 cycles of 95 °C for 3 s, 60 °C for 30 s. For each experiment, a standard curve was calculated using 6 standards ranging from $10^2$ to $10^7$ genome copies per reaction amplified in duplicate on each 96-well plate. Virus genome copy number per reaction was determined by absolute quantification using a standard curve, with the lower limit of detection set at the lowest standard ($10^2$ genome copies per reaction). Viral copy numbers in LP control samples were compared between individual infection experiments using Kruskal–Wallis and a Dunn's post-test to ensure that infection data could be included in pooled experiments. All individual experiments are displayed in Supplementary Fig. S2.

### Hemocytometer assays to determine hemocyte percentages
To quantify the efficacy of CLD depletion on phagocytic hemocytes, mosquitoes (3–5 days old) were injected with 69 nl of 1:4 diluted LP or CLD as described above. Surviving mosquitoes were challenged with defibrinated sheep blood (Hemostat Laboratory) 24 h post-injection. Hemolymph was perfused at 1, 4, 7, and 10 days post-blood feeding using an anticoagulant solution as previously described[39,46,76]. The proportion of granulocytes was calculated by analyzing ~200 hemocytes per individual mosquito based on morphology using a disposable Neubauer hemocytometer slide (C-Chip DHC-N01; INCYTO) as previously described[39,46,76].

### In vitro virus infection assays in the presence of clodronate
To determine if the observed reductions in virus infections following hemocyte depletion were not a side effect of clodronate antiviral activity, we examined DENV and ZIKV growth kinetics in the presence of clodronate in vitro. Briefly, C6/36 cells were plated in 12-well plates at a density of $1 \times 10^6$ cells/well. Once the cells were confluent, DENV2 or ZIKV stock was diluted in 10% DMEM in the presence of clodronate disodium salt (Sigma-Aldrich; 0 μM, 10 μM, 20 μM, and 40 μM) and cells infected at an MOI of 0.1 for 1 hr. at RT with gentle rocking. Subsequently, virus was removed, the cells washed twice with PBS, and 1 ml of fresh media containing 0 μM, 10 μM, 20 μM, and 40 μM of clodronate salt added back to the wells. 250 μl samples were collected at 24, 48, 72, and 120 hpi. Total nucleic acid was extracted from 200 μl using the MagMax Viral/Pathogen Nucleic Acid Isolation kit (Thermo Fisher) using a Kingfisher Flex high-throughput extraction device and eluted in 50 μl. DENV and ZIKV genome equivalents were determined by RT-qPCR as described above.

### Mosquito survival following blood-feeding and infection
To evaluate the effect of phagocytic hemocyte depletion on mosquito survival following blood-feeding and virus infection, mosquitoes were treated with either LP or CLD at 1:4 dilution as described above, and then challenged 1-day post-treatment with an artificial blood meal containing non-infected sheep blood, or infected blood containing DENV or ZIKV. After challenge, mosquitoes that were fully engorged were selected and then monitored daily over a 10-day period to assess survival. Experiments were replicated in three independent biological replicates, with ~50–60 individual mosquitoes per treatment in each experimental replicate.

### Immunofluorescence assays for localization of viruses in hemocytes
Hemocyte immunofluorescence assays (IFAs) were performed as previously described[39,46,77]. Following infection with DENV, mosquitoes were injected with 69 nl of a 2% green fluorescent FluoSpheres solution (in 1X PBS) using a Nanoject III (Drummond Scientific) and incubated at 27 °C for 1 h to allow for bead uptake by hemocytes at 10 days post-infection. Hemolymph perfusion was carried out as previously described[39,42,46,77,78]. Briefly, an incision was made in the second to last abdominal segment, and ~10 μl of anticoagulant solution (60% [vol/vol]

Schneider's insect medium, 10% FBS, and 30% citrate buffer; 98 mM NaOH, 186 mM NaCl, 1.7 mM EDTA, and 41 mM citric acid, pH 4.5) was used to perfuse hemolymph using a Nanoject III injector onto a multi-well glass slide (MP Biomedicals). After 30 min incubation, cells were fixed with 4% paraformaldehyde for 15 min at room temperature (RT) and then washed three times in 1X PBS. Samples were incubated with blocking buffer (0.1% Triton X-100, 1% BSA in 1X PBS) for 24 h at 4 °C, then with an anti-DENV monoclonal antibody (clone 3H5-1; BEI Resources) at a 1:200 dilution in blocking buffer overnight at 4 °C. After washing three times in 1X PBS, an Alexa Fluor 568 goat anti-mouse IgG (1:500, Thermo Fisher Scientific) secondary antibody in blocking buffer was added for 2 h at RT. Slides were rinsed three times in 1X PBS and mounted with ProLong®Diamond Antifade mountant with DAPI (Life Technologies). Images were examined by fluorescence microscopy (Nikon Eclipse 50i, Nikon). To determine the proportion of phagocytic hemocytes infected by virus, 50 hemocytes from randomly chosen fields were examined for the presence or absence of virus signal and/or beads.

### Imaging of hemocyte attachment to mosquito salivary glands and ovaries
At 7 days post-DENV or -ZIKV infection, mosquitoes were injected with 69 nl of a solution containing 100 μM Vybrant CM-DiI (ThermoFisher Scientific) and 2% green fluorescent beads in 1X PBS, then incubated under insectary conditions for 30 min. To preserve hemocyte attachment, mosquitoes were injected with 207 nl of 50% ethanol in 1X PBS. Salivary glands and ovaries were then dissected in 1X PBS and mounted using ProLong®Diamond Antifade Mountant with DAPI. The attachment of phagocytic granulocyte populations (cells that are DiI+/bead+) to salivary glands and ovaries were examined by fluorescence microscopy (Nikon Eclipse 50i, Nikon).

### Quantification of hemocyte attachment to gut, ovaries, and salivary glands
To examine whether bead injection induces hemocyte attachment to the midgut, ovaries, or salivary glands, Naïve mosquitoes were injected with 69 nl of a solution containing 100 μM Vybrant CM-DiI (Thermo Fisher Scientific), 1 mM Hoechst 33342 (Thermo Fisher Scientific), and 2% green fluorescent beads in 1X PBS, then incubated under insectary conditions for 30 min. To preserve hemocyte attachment, mosquitoes were injected with 207 nl of 50% ethanol in 1X PBS. Mosquito midguts, salivary glands, and ovaries were then immediately dissected in cold 1X PBS and mounted in Aqua-Poly/Mount (Polysciences). The number of tissue-associated hemocytes (Attached DiI+ cells) were then quantified by fluorescence microscopy.

To quantify hemocyte attachment to the midgut, ovaries, and salivary glands over time and under different infection conditions, mosquitoes were either infected orally with DENV or fed a 1:1 mixture of defibrinated sheep's blood and L-15 media supplemented with 2% FBS and 4 mM L-glutamine (blood-fed). Hemocyte attachment was quantified for midguts, salivary glands, and ovaries collected at 3, 7, and 10 days post-infection or post-blood feeding. Hemocyte attachment was also quantified for age-matched naïve mosquitoes before infection and at each time point examined. Before dissection, mosquitoes were injected with 69 nl of a solution containing 100 μM Vybrant CM-DiI and 1 mM Hoechst 33342 in 1X PBS, then incubated under insectary conditions for 30 min. Due to previous experiments demonstrating significantly increased hemocyte attachment after bead injection, we chose to omit fluorescent beads from these experiments. To preserve hemocyte attachment, mosquitoes were injected with 207 nl of 50% ethanol in 1X PBS. Mosquito midguts, salivary glands, and ovaries were then immediately dissected in cold 1X PBS and mounted in Aqua-Poly/Mount. The number of tissue-associated hemocytes (Attached DiI+ cells) were then quantified by fluorescence microscopy.

## Virus dissemination experiments

To assess the impact of hemocyte depletion on virus dissemination, mosquitoes were first orally challenged with either DENV or ZIKV, then injected with 69 nl of a 1:4 dilution of LP or CLD as previously described[40] at 3 days post-infection. To assess DENV dissemination, mosquito legs, ovaries, and salivary glands were harvested at 8-, 10-, and 12-days post-infection. For ZIKV assays, mosquito legs, ovaries, and salivary glands were harvested at 8- and 10-days post-infection. RNA was isolated from dissected tissues with a Mag-Bind Viral DNA/RNA 96 Kit (Omega Bio-tek Inc) and Kingfisher Flex automated nucleic acid extraction device (ThermoFisher Scientific) per the manufacturer's instructions. Viral RNA titers and infections were assessed via qRT-PCR using an iTaq Universal Probes One-Step Kit (BioRad) in a 20 µl sample volume. Reactions were performed using previously described primers for DENV-2[79] or ZIKV[75] (Supplementary Table S3) with the following PCR conditions: 50 °C for 30 min, 95 °C for 10 min and 40 rounds of 50 °C for 15 s and 60 °C for 1 min. Viral RNA titers were quantified using a standard curve with a cut-off value of Ct < 36. Sample Ct values below this cut-off were considered positive whereas samples with larger Ct values were considered negative. Viral concentrations were corrected for dilution factor. Results were pooled for DENV and ZIKV assays from either three or two independent biological replicates respectively.

## Hemolymph perfusion and hemocyte transfer experiments

To examine the infectivity of the cellular and acellular fractions of the mosquito hemolymph, cellular and cell-free hemolymph fractions were isolated from virus-infected mosquitoes and transferred to naïve mosquitoes similar to previous studies[50,51]. At 10 days post-infection with DENV or ZIKV, perfused hemolymph was collected using established injection-recovery protocols for hemolymph perfusion[39,46,50,78] from individual mosquitoes using ~8–10 µl of transfer buffer (95% Schneider's medium and 5% citrate buffer). Approximately 5 µl of buffer/hemolymph were collected for each mosquito, with hemolymph (1–2 µl/mosquito) estimated to comprise ~10–20% of the recovered sample. Pooled samples ($n = 15$) underwent centrifugation at 4 °C, 2000 × $g$ for 5 min to separate cell-free supernatant (SUP) and hemocyte-containing (CELL) fractions. The resulting SUP fraction was transferred to a new tube and used directly in downstream quantification or transfer experiments. The CELL fraction was washed by resuspending in 1X PBS and again centrifuged at 4 °C, 2000 × $g$ for 5 min. Following centrifugation, the 1X PBS was removed, with the resulting CELL pellet resuspended in an equivalent volume to that of the SUP fraction using transfer buffer. For transfer, naïve mosquitoes were injected with 207 nl of either the prepared SUP or CELL fractions. Whole mosquitoes were collected at 1, 2, and 4 days post-injection for DENV and 4 days post-injection for ZIKV. Whole mosquitoes were homogenized in 400 µl TRIzol Reagent and RNA isolation was then completed using Direct-zol RNA MiniPrep kit. Viral copy numbers of whole mosquitoes were analyzed by qRT-PCR using the above described protocols for midgut infection experiments using 100 ng of template RNA per reaction.

Additional experiments were performed to examine the effects of phagocyte depletion on the infectivity of cellular and acellular hemolymph fractions. Briefly, naïve mosquitoes were injected with 69 nl of LP or CLD (1:4 dilution) as described above, then challenged with an oral infection of DENV or ZIKV one day later. At 10 days post-infection, hemolymph perfusion, isolation of the SUP and CELL fractions, and transfer to naïve mosquitoes was carried out as described above. Viral RNA titers of whole mosquitoes were examined by qRT-PCR 4 days post-transfer.

To assess which tissues become infected after the transfer of infected hemocytes, DENV or ZIKV-infected mosquitoes were perfused at 10 days post-infection, and the perfusate was separated into SUP and CELL fractions as described above. Naïve mosquitoes were then injected with 207 nl of the prepared CELL fraction. At 4 days post-injection, salivary gland, ovary, midgut, and carcass tissues were isolated in cold 1X PBS. Viral RNA levels for each tissue sample were determined by qRT-PCR, and a subset of salivary gland and ovary samples were analyzed instead by FFA to confirm the presence of infectious virus.

To confirm the infectivity of the cellular and acellular hemolymph fractions in vitro, C6/36 cells were first seeded into a 24-well plate at ~$1 × 10^6$ cells per well and allowed to attach for 24 h. Collected SUP and CELL fractions were diluted in 600 µl of L-15 media supplemented with 2% FBS, 4 mM L-glutamine, 100 units/ml penicillin, 100 µg/ml streptomycin and 0.25 µg/ml amphotericin B. Then 300 µl of infectious media from each fraction was added to C6/36 cells in duplicate. To determine the initial titer of each hemolymph fraction, additional SUP and CELL isolates collected from the same group of infected mosquitoes were directly lysed in 400 µl TRIzol Reagent and left at 4 °C overnight to maximize yield. Infection outcomes were examined at 2- and 3-days post-infection by resuspending the C6/36 cells by vigorously pipetting up and down, then 100 µl of the cell suspension was added to 900 µl TRIzol Reagent. RNA isolation was then completed using Direct-zol RNA MiniPrep kit and viral RNA titers determined by qRT-PCR.

## Focus forming assays

To assess viral titer by Focus Forming Assay (FFA), 96-well plates were seeded with C6/36 cells at a density of $3 × 10^5$ cells per well and incubated at 28 °C overnight. FFAs were performed at ISU and CAES with slight modification. At ISU, mosquito samples (SUP/CELL fractions and individual tissues) were homogenized in L-15 media supplemented with 2% FBS, 4 mM L-glutamine, 100 units/ml penicillin, 100 µg/ml streptomycin, and 0.25 µg/ml amphotericin B. Homogenized samples were then serially diluted and 30 µl of each dilution was plated per well. Cells were infected for 1 h, then infectious media was removed and replaced with 100 µl 1% methylcellulose in L-15 media supplemented with 2% FBS and 4 mM L-glutamine and incubated at 28 °C for 4 days. Cells were then fixed for 15 min at room temperature with 4% paraformaldehyde in 1X PBS, washed 3 times with 100 µl 1X PBS, and permeabilized with 100 µl blocking buffer at 4 °C overnight. At CAES, samples (virus-infected blood and tissue samples) were processed as previously described[19], with dilutions performed using serum-free minimum essential medium with samples homogenized in 1X PBS. Methylcellulose (1%) was suspended in minimum essential medium containing 2% fetal bovine serum. Cells were fixed for 15 min at room temperature with 100 µl of 4% formaldehyde in 1X PBS, washed 3 times with 100 µl 1X PBS, and permeabilized with 0.2% Triton-X in 1X PBS for 10 min at room temperature. At both locations, foci were labeled with 30 µl of mouse anti-flavivirus group antibody (D1-4G2-4-15) diluted at 1:250 (Millipore Sigma MAB10216) or at 1:500 (NovusBio NBP2-52709) in 1X PBS and incubated at 4 °C overnight. Cells were then washed 3 times in 1X PBS and 30 µl of goat anti-mouse IgG (H + L) cross-absorbed Alexa Fluor 488 (1:200) or Alexa Fluor 568 (1:250) secondary antibody was diluted in 1X PBS or blocking buffer. Cells were incubated for 2 h at room temperature or overnight at 4 °C. Plates were washed 2 times in 1X PBS, washed a final time in de-ionized water, and allowed to dry. Foci were quantified using fluorescence microscopy.

## Software

All data analysis and visualization were performed using GraphPad Prism (version 10.4.2). Figures were created using Inkscape (version 1.0.2-2). Additional schematic illustrations were created in Inkscape and supplemented using images developed with BioRender (BioRender 2024).

## Reporting summary

Further information on research design is available in the Nature Portfolio Reporting Summary linked to this article.

## Data availability

All source data supporting the findings of this study are included in the manuscript and provided in the Supplementary Information as Source Data files. Source data are provided with this paper.

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

## Acknowledgements

This work was supported by R21 AI149118 (DEB and RCS) and R01 AI148477 (DEB) from the National Institutes of Health, National Institute of Allergy and Infectious Diseases. This material is based upon work supported by the National Science Foundation Graduate Research Fellowship Program under Grant No. 2336877. Any opinions, findings, and conclusions or recommendations expressed in this material are those of the authors and do not necessarily reflect the views of the National Science Foundation. The monoclonal anti-dengue virus type 2 envelope

protein antibody, clone 3H5-1 (NR-2556) was obtained through BEI Resources, NIAID, NIH.

## Author contributions

D.R.H., R.M.J., and H.K. contributed equally and are listed in alphabetical order. D.R.H., R.M.J., H.K., D.E.B., and R.C.S. conceived the study. D.R.H., R.M.J., H.K., Z.F. and D.E.B. performed experiments. S.V.L.-T. and B.J.B. contributed reagents and assisted in virus culture. B.J.B., D.E.B., and R.C.S. provided supervision and experimental oversight. D.R.H. and R.C.S. wrote the initial draft of the manuscript. All authors reviewed and edited the initial manuscript. All authors approved the final manuscript.

## Competing interests

The authors declare no competing interests.
