## [Transparent Peer Review file · Nature Communications]

Mosquito immune cells enhance dengue and Zika virus infection in *Aedes aegypti*

Corresponding Author: Dr Ryan Smith

Version 0:

Reviewer comments:

Reviewer #1

(Remarks to the Author)

In this paper, Hall et al analyse the role of hemocytes, the immune cells of insects, in arboviral dissemination – ZIKV and DENV- in the mosquito *Aedes aegypti*.

So far, the role of hemocytes (pro or antiviral) in arbovirus transmission by mosquitoes has remained unknown due to the lack of genetic tools allowing to assess this question. Using elegant experiments, the authors show that :

Granulocyte depletion does not influence midgut DENV or ZIKV titers

Virus primarily infects phagocytic granulocyte populations of mosquito hemocytes

Phagocytic granulocytes adhere to tissues involved in arbovirus transmission

Granulocyte depletion attenuates virus dissemination

Phagocytic granulocytes transmit virus to uninfected tissues

While this study of great interest, some important conclusions, including the title, are overstated in the absence of supplementary controls. Indeed, the conclusion that they demonstrate that phagocytic hemocytes ... facilitate virus dissemination to important tissues such as the salivary glands and ovaries in the mosquito *Ae. aegypti* (lines 212-214, 269, 320, etc), is not supported by the data. While the data indeed suggest it could be, there is no clear demonstration but only correlations of different data, which themselves can be criticised. As detailed below, to make the demonstration, the authors should demonstrate that CLD alone does not affect virus viability and show that the transfer of infected hemocytes well infect tissues of naïve mosquitoes (not just whole mosquitoes), and this by virus titration.

Major modifications

While clodronate has been shown to deplete granulocytes, what is the effect of clodronate on virus alone or virus replication? We cannot exclude that it is antiviral, and could explain the decrease of midgut infection (rather than an indirect role of hemocytes increasing midgut infection). In the absence of an experiment showing that this compound has no direct antiviral effect, the conclusion that hemocyte depletion decrease virus dissemination is not supported. This is also relevant for the experiment with transfer of supernatant or cell (figure 5e and f). Especially at the light of supernatant from LP treated ZIKV mosquitoes giving rise to some infection in transferred mosquitoes while not in the CLD ones.

Replicates were pooled. How was it performed. The possibility to combine replicates should be statistically tested to make sure there is not an effect of the experiment.

Diff percentage granulocytes between figs1 and fig 2b? How is this explained? I wonder why the authors do not use the beads to identify and count them (as in fig 2). Could the beads induce their differentiation into phagocytic granulocytes explaining why the percentage is lower in figS1? This should be addressed.

Since the authors observed an effect on virus prevalence in the gut (figure 1), they then bypass the influence of phagocyte depletion on midgut infection by injecting clodronate liposomes three days post-infection. While they looked at dissemination in different tissues, they did not check for the midgut infection. Flaviviruses takes a long time to disseminate from the gut, more than 3 days. In the absence of measure of infection in the midgut showing no difference in infection titers and prevalence, the conclusion that there is less dissemination after depletion cannot be supported.

Figure 4: titres, and not only prevalence, for all tissues should be shown.

When transferring supernatant or hemocytes (fig 5), the authors measure by qPCR the RNA viral loads in whole mosquitoes. First, this is not a measure of transmission to other tissues, could just be the initial infected hemocytes transferred.

Hemocytes are known to divide, so the increase in virus load by qPCR seen with DENV may not be infection of mosquito tissues but an increase of hemocyte infection. Second, while viral loads measured by qPCR are more and more accepted although some will still not accept this as proof of infection (as opposed to virus titration, ie live virus), for this particular

transfer experiment, I would have liked to see virus titration to make sure the virus transferred is still live and not just genome traces, especially in the case of ZIKV where only one time point was looked.

The fact that virus collected from the supernatant fraction resulted in a significantly attenuated infection, and to be honest, it is not attenuated, two data points show clearly no infection at all, puts into question the data obtained with DENV, regarding the fact that cell free virus in hemolymph cannot participate to dissemination. Virus inactivation when not in a cell is rapid and the fact that the virus is inactivated after hemolymph perfusion, centrifugation does not necessarily mean the cell free virus does not contribute to dissemination within one mosquito.

I recommend the authors to stay prudent with conclusions about cell free virus and do not overstate that the hemocytes are the primary component of mosquito hemolymph that is able to promote virus dissemination. Hemolymph free virus may contribute, and this does not remove the interesting data that hemocytes also do. Regarding hemocytes, I'd like to see more controls, such as virus titration of different tissues, and not just whole mosquitoes. Otherwise, the conclusion that granulocytes are capable of disseminating viruses to mosquito tissues (line 203-204 for example) is invalid.

Minor modifications

L94: CLD treatment may still be efficient thanks to liposome presence after 10 days. In addition, the study looks up to 10 days, therefore, I would recommend staying cautious with the statement that "phagocyte depletion is permanent for the life of the mosquito", which can live up to a month or so. Rephrase, eg phagocyte depletion was efficient at least up to 10 days.

L100: viral titers by qRT-PCR > viral loads (titres are obtained with titration, not qPCR) – To check all manuscript including legends.

References: Virus infected by arboviruses (alphavirus ONNV) and hemocytes attached to tissues, including midgut and trachea, please cite Hemocyte-targeted gene expression in the female malaria mosquito using the hemolymph promoter from *Drosophila* - ScienceDirect

Reviewer #2

(Remarks to the Author)

This manuscript demonstrates the role that hemocytes play in the dissemination of two medically important flaviviruses within the hemocoel of the mosquito vector. The data are significant because hemocytes are considered protective against infection, yet the demonstration here is that they can also be a liability, presumably affecting vectorial capacity. Overall, I am supportive of the manuscript but below highlight some important points.

1. Granulocyte percentage and identification in figures S1 and 2. Figure S1 shows that CLD treatment decreases the percentage of granulocytes, and places granulocytes as 8-15% of the hemocyte population. Yet, in figure 2a the granulocytes are prevalent in the image; 8 of the 9 cells (89%) are phagocytic and presumably granulocytes. In figure 2b, the percentage of granulocytes is quantified as 90% or greater (add the first two columns). How can this contradiction be reconciled? Also, can the cell types in figure 2a be labeled?
2. Figure 3. Frequency of occurrence matters. Figure 3 demonstrates that hemocytes can attach to the salivary glands and ovaries, which is relevant to the scientific question because these are important destinations of the virus. However, the frequency of these occurrences is not mentioned. Are hemocytes most often seen attached to these tissues, or do they attach seldomly? Are there usually many hemocytes at these locations, or are they usually few?
3. Figure 4. Magnitude matters. The findings in this figure are presented in binary terms. Treatment either affects or does not affect. More nuance is needed. For example, in figure 2b all timepoints are significantly different, but the differences at the later two timepoints are small. Likewise, the prevalence phenotypes in figure 2C are significantly different in some timepoints but not in others. For transparency, the narrative should capture the nuances.
4. Hemolymph transfusion experiments, part 1 (Fig 5a-c; start in line 162). These experiments are very interesting, and difficult. In reading the manuscript, I had difficulty conceptualizing some of the findings, primarily because the hemolymph being used is collected via perfusion. In other words, it is not pure hemolymph, and in fact, the majority of the fluid is the transfer buffer and not hemolymph. More details on the methodology are needed. Hemolymph is estimated to be what percentage of the collection? Is the CELL component more concentrated than the SUP component? Was the cell component microscopically visualized to see whether granulocytes are there? How can it be explained that viral titer in CELL and SUP are similar yet there are some differences in how they infect cells in vitro?
5. Hemolymph transfusion experiments, part 2 (Fig 5d-f; start in line 192). Could it be that the lower infectivity when hemolymph from CLD mosquitoes is transfused is simply because dissemination has been lower (see fig 3).
6. In the discussion, the authors tackle two potential discrepancies between their data and the data of others. The discrepancy outlined starting in line 215 makes complete sense, and I agree with the authors (I would also expect phagocytosis saturation and CLD treatment to yield different outcomes). However, I am less convinced regarding the discrepancy that starts in line 230. Cheng et al 2022 claim that prohemocytes are the virus infected cell whereas the present manuscript states that granulocytes are the virus infected cell. I do not know the answer, and I see problems with the claims in both studies. As pertains to the manuscript being evaluated here, the authors demonstrate that granulocytes are infected, but there isn't any convincing evidence that prohemocytes are not. Given that Fig S1 shows that 8-15% of the hemocyte population are granulocytes and fig 2 focuses only on granulocytes, even a small percentage of prohemocytes (presumably most of the remaining 85-92%) infected could dwarf granulocyte infection.
7. It would have been highly informative to see data on mosquito survival. Does CLD treatment cause more mosquitoes to

die when infected with a virus? If so, this would suggest that hemocytes enhance dissemination while protecting the mosquito's reproductive success. In other words, from the perspective of the mosquito, the hemocytes would still be beneficial.

8. One final thought that the authors should consider is that the importance of granulocyte infection could be more so as replication factories and less so as dissemination vehicles. This does not challenge anything in the manuscript, but I mention it because the dynamic nature of hemolymph flow might mean that, inside hemocytes, replication may be more important than transportation.

Reviewer #3

(Remarks to the Author)

The article by Hall et al addresses an important question about the role of circulating immune cells during arbovirus infection in mosquitoes. The major hypothesis in the manuscript is that mosquito phagocytes acquire the arboviruses in the mosquito gut and then are essential to disseminate these viruses to other tissues, such as the salivary gland. This is an interesting hypothesis that has been proposed before, but the article does not present experiments that directly prove it. Basically, the manuscript has experiments that establish a correlation. First, authors show that hemocytes are infected by dengue and Zika viruses and find infected hemocytes adhering to the ovaries and salivary glands. However, this is a correlation and not proof that these hemocytes were the vessel for virus dissemination. It is possible that hemocytes might have migrated later to assist in the immune response against infection or to deal with tissue damage. Second, authors show that transference of infected hemocytes is very efficient at infecting new mosquitoes but not cell-free viruses. This is the highlight of this manuscript since most of the results presented are not new. For example, the pro-viral role of hemocytes during dengue and Zika virus infections has been demonstrated (PMIDs: 34093550) as well as the role of hemocytes as a target for arbovirus infections (PMIDs: 34093550, 25548172, 17263893, 19141437). However, the infectiousness of cell associated virus does not prove that hemocytes themselves are the vessel for virus dissemination from the mosquito gut to other tissues. I think this observation could be the basis for an entire manuscript to be developed but, at the moment, it lacks mechanistic understanding. Hence, I think this manuscript would be a better fit for a more specialized journal.

Specific comments:

In figure 1 there is a clear trend of lower infection in the midgut when hemocytes are depleted, as previously reported when phagocytosis is blocked (PMIDs: 34093550). This is an important point that needs to be addressed, especially since the authors use clodronate in this manuscript compared to latex beads in Leite et al (PMIDs: 34093550). Based on the supplementary data, the experiment was performed three times. Does the figure show all of them combined? How do each of them separately look like? Overall, the authors have a model that suggests saturating infection, with over 80% of infected mosquitoes, which could mask any effects of hemocytes in the midgut. Thus I disagree that there is no effect on midgut infection when hemocytes are depleted. Authors could have looked at earlier time points and analyzed direct midgut infection by imaging as they did for other organs.

Figure 2 shows data for 10 days post infection – how representative would this be during the kinetics of the infection? According to the major hypothesis, hemocytes should be infected early. Here it would also be good to use the same dye and in figure 3, since beads could affect hemocyte physiology.

In figure 3, in order to have more meaningful information, the number of hemocytes attached to salivary glands and ovaries should be quantified during the kinetics of viral infection. This should be compared to the infection in these organs as compared to the recruitment of hemocytes to establish the idea that hemocytes arrive before the infection becomes clear. The presence of hemocytes in these organs should also be analyzed in the absence of infection. It has been shown that injection of beads can increase the amount of hemocytes associated to the gut (PMIDs: 34093550) and this may be true for other tissues. Thus, it is important to have a control group without beads.

In figure 4, viral titers should be shown for all results. Delaying dissemination or directly hosting viral replication would basically translate into the same effect and the authors cannot exclude the two possibilities. Furthermore, without knowing whether there is a clear effect on midgut infection, it is hard to analyze anything after, since dissemination depends on infection at the primary site.

The results in figure 5 are the highlight of the manuscript, but they do not prove the authors hypothesis. Results that cell associated viruses are more infectious to new mosquitoes do not prove that viruses are disseminated within hemocytes in the infected mosquito. Furthermore, authors show that dengue in the supernatant is less viable than in cell fractions, although not true for Zika, which deserves further analysis. For example, is it a matter of timing? If they were to analyze an earlier time point with Zika, would it be different? Overall, the observation is poorly explored by the authors. Supplementary figure 3, that complements these results, should be integrated into a main figure. More experiments are required to help explain these results.

In figure S2, statistics on 2 replicates should be removed.

Version 1:

Reviewer comments:

Reviewer #1

(Remarks to the Author)

Every point raised has been addressed.

Congratulations to the authors for this excellent paper!

Reviewer #2

(Remarks to the Author)

This is an improved manuscript demonstrating that dengue and Zika viruses can replicate in granulocytes to amplify the infection, and that these infected granulocytes can be a vehicle for the artificial transmission of viruses from one mosquito to another. The body of work is impressive, and the experiments are well designed and conducted. The findings are significant, but my main concern is the overinterpretation of the meaning of the data. I will focus on two main points:

1. In the rebuttal, the investigators explain that the difference in the cell type percentages between figure panels (granulocytes versus prohemocytes versus oenocytoids) is that different methods yield different outcomes: using a hemocytometer, granulocytes are the smallest population, but using microscope slides they are the largest population. The claim is made that the hemocytometer provides the true proportion. The experiment where the authors conclude that granulocytes are the main cells that becomes infected used the microscope slide method. Using this method, the authors note that 70-95% of cells are lost (most of the prohemocytes and oenocytoids), so without the other cells present it seems likely that granulocytes would be the most infected. Therefore, can it really be claimed that granulocytes are the predominant infected cell when the method dramatically enriches this population? This is especially the case because it is not noted whether prohemocytes (which have some phagocytic activity) or oenocytoids are depleted by clodronate. For that reason, an unsupported claim is that "These results provide strong evidence that phagocytic granulocytes comprise the majority of virus-infected hemocytes and indicate that phagocytic granulocyte populations may be important for virus dissemination".

2. Another primary claim is that hemocytes disseminate the infection. A clear distinction between replication and dissemination should be made. Replication means that viral copy number increases, and the manuscript makes a very convincing argument that this is the case. Dissemination means that the virus is transported, and this claim is significantly weaker. Hemocytes attach to all tissues, and surely their attachment to the salivary glands brings the virus closer to where they need to be for transmission. But this does not discount the possibility that soluble virus is infecting the salivary glands (or other tissues). The claim for dissemination is made using two lines of evidence. The first is that CLD treatment reduces virus dissemination to the legs. The experiment demonstrates that there is a delay in dissemination but not that the hemocytes are disseminating (other than by being replication factories). The second is that injection of the cellular portion of hemolymph causes infection in naïve mosquitoes but injection of cell free hemolymph does not. The problem here is that the cellular fraction is concentrated by centrifugation (so it may be more concentrated than actual hemolymph) but the cell free fraction is significantly diluted by the solution used to perfuse. So, the experimental design shows that infected hemocytes can infect, but the statement that the cell free fraction is not infective is weak.

Reviewer #3

(Remarks to the Author)

The revised article by Hall et al titled "Mosquito immune cells enhance dengue and Zika virus dissemination in *Aedes aegypti*" has improved over the previous version but some key questions remain. I appreciate the efforts from the authors but, overall, the data do not fully support the hypothesis that hemocytes work as trojan horses to disseminate dengue and Zika viruses. Without further mechanistic insights and significant new data, I think the manuscript is more appropriate for a specialized journal.

Below are my major criticisms:

The data on the kinetics of hemocyte infection (Figure 2), hemocyte association with tissues (Figure 3) and tissue infection (Figure 4) together show a scenario that does not fit the idea that hemocytes are themselves the vehicle for virus dissemination. First, hemocytes clearly become infected at the same time or even later than other tissues. Second, how do the authors address the matter that hemocyte association with tissues is not affected by the infection? How do hemocytes help spread the virus if they are always associated with the tissue? If they are already associated with tissues such as ovaries and salivary glands at 3 days post feeding to the same levels but only become infected at 6 dpf, how do the authors suggest they spread the virus. This would require a lot of re-localization of hemocytes. Authors should show infected hemocytes in the tissue and not only cells obtained after perfusion. Third, the levels of tissue associated hemocytes does not match the idea they drive virus dissemination. There are at least 5 times more hemocytes associated with ovaries than salivary glands in their dataset but, yet, infection is higher in salivary glands. How to account for these inconsistencies?

I reinforce that the preliminary data shown in Figure 5 are the highlight of the manuscript that can be developed into something interesting. However, authors have not addressed my previous comments and no new further data was included.

Finally, I also have two technical concerns:

1) The experiment described in Figure 4 shows a marginal effect on viral loads measured by RT-qPCR but a significant effect on prevalence. However, that is not reproduced when the authors look at virus titrations (Supplementary Figure S6). What is their explanation?

2) RT-qPCR to detect virus in RNA extracted from dissected tissues does not show active replication in the tissue as the authors mention in their rebuttal. In contrast, IFA of the tissue is much better at determining where the virus is replicating.

Version 2:

Reviewer comments:

Reviewer #2

(Remarks to the Author)

The authors have addressed my comments and I am supportive of the manuscript.

Reviewer #3

(Remarks to the Author)

This newly revised manuscript by Hall et al did not provide any new compelling evidence to change my opinion about the work. I appreciate the carefully crafted response but, I have to repeat myself that without further mechanistic insights and significant new data, I think the manuscript is more appropriate for a specialized journal. On many of the specific points, I agree with the authors but they did not address the major concerns. It is true that there is not a lot of information about the contribution of hemocytes to arbovirus infections in mosquitoes. While this means that many questions remain unanswered, the work by Hall et al did not bring a significant contribution to the subject. The message that hemocytes have a proviral role during dengue and Zika virus infections is not novel. I reinforce that data in figure 5 should be the main focus of the work but it is just an initial basis for the hypothesis that hemocytes are the vessels for virus dissemination. I appreciate the response from the authors but it is not enough, more substantive data is required to support the hypothesis.

Response to Reviewer's comments

NCOMMS-24-24225-T

"Mosquito immune cells enhance dengue and Zika virus dissemination in *Aedes aegypti*"

A detailed response to each of the reviewer comments is listed below. All changes in response to the reviewer's comments are highlighted with the "tracking changes" function in the manuscript text.

Reviewer #1

While this study of great interest, some important conclusions, including the title, are overstated in the absence of supplementary controls. Indeed, the conclusion that they demonstrate that phagocytic hemocytes ... facilitate virus dissemination to important tissues such as the salivary glands and ovaries in the mosquito *Ae. aegypti* (lines 212-214, 269, 320, etc), is not supported by the data. While the data indeed suggest it could be, there is no clear demonstration but only correlations of different data, which themselves can be criticised. As detailed below, to make the demonstration, the authors should demonstrate that CLD alone does not affect virus viability and show that the transfer of infected hemocytes well infect tissues of naïve mosquitoes (not just whole mosquitoes), and this by virus titration.

We would like to thank the reviewer for their thoughtful review. Through additional experiments we believe that we have addressed your previous reservations in our revised manuscript, including virus infection experiments in the presence of clodronate, additional analysis of tissues following hemocyte transfer, and the titration of virus using focus-forming assays. Additional details are provided for each of the comments below.

Reviewer Comments

1. While clodronate has been shown to deplete granulocytes, what is the effect of clodronate on virus alone or virus replication? We cannot exclude that it is antiviral, and could explain the decrease of midgut infection (rather than an indirect role of hemocytes increasing midgut infection). In the absence of an experiment showing that this compound has no direct antiviral effect, the conclusion that hemocyte depletion decrease virus dissemination is not supported. This is also relevant for the experiment with transfer of supernatant or cell (figure 5e and f). Especially at the light of supernatant from LP treated ZIKV mosquitoes giving rise to some infection in transferred mosquitoes while not in the CLD ones.

We would like to thank the reviewer for the suggestion. In our revised manuscript we have performed additional experiments using *in vitro* cell culture to examine the potential that clodronate could interfere with DENV or ZIKV replication. In these experiments, we use clodronate disodium salt, the compound encapsulated in clodronate liposomes, to evaluate the potential effects of clodronate at three different concentrations on virus infection using C6/36 cells at 24, 48, 72, and 120 hpi.

As expected, exposure to clodronate had no effect on DENV or ZIKV *in vitro*. For both viruses, the presence of clodronate did not impair virus infection (as indicated by the 24 hpi timepoint), nor did clodronate influence virus replication as measured by the 48, 72, and 120 hpi timepoints. These data are now included as Fig. 1d-f, as part of a revised Fig. 1.

Without liposomes as a delivery mechanism, non-encapsulated clodronate cannot cross the cell membrane to initiate cell death. These experiments demonstrate that clodronate does not interfere with the ability of a virus to infect or further replicate in a host cell, further supporting that the observed phenotypes are the results of phagocyte (granulocyte) depletion.

2. Replicates were pooled. How was it performed. The possibility to combine replicates should be statistically tested to make sure there is not an effect of the experiment.

In response to the reviewer comment, we have now included midgut infection data (virus copies/prevalence) for each of the individual experiments as part of Fig. S2 in our revised manuscript. In our initial submission, we did have one DENV midgut infection replicate that was statistically different (Kruskal-Wallis with Dunn’s post-test of virus copies in LP control samples). In our resubmission, we have now removed this from the pooled data presented in Fig 1b, yet we still display data from this experimental replicate in Fig. S2. Since we did exclude an experiment from this analysis, we did perform another independent experiment (Exp. #5) for our DENV midgut infection data. The LP controls for Exp. #5 were not significantly different from the other experiments and were subsequently included in a revised Fig. 1b. Data from each of the five experiments are displayed in Fig. S2, with additional modification in the methods and figure legend text to reflect these changes in the revised manuscript.

3. Diff percentage granulocytes between figs1 and fig 2b? How is this explained? I wonder why the authors do not use the beads to identify and count them (as in fig 2). Could the beads induce their differentiation into phagocytic granulocytes explaining why the percentage is lower in figs1? This should be addressed.

The percentage of granulocytes in Fig. S1 and Fig. 2 were evaluated through different previously established methodologies for examining mosquito hemocyte populations. Data presented in Fig. S1 were performed using hemocytometer assays, while those in Fig. 2 were performed via immunofluorescence using fixed hemocytes on a slide. As described in Kwon and Smith (2019- PNAS), these methods (represented here in A and B in Figure S2 from this paper below) result in different percentages of granulocytes.

Methodology	Proportion of hemocytes			References
A Hemocytometer 	Prohemocytes  ~40-60%	Oenocytoids  ~30-50%	Granulocytes  ~5-30%	Rodrigues et al. 2010 Garver et al. 2013 Ramirez et al. 2014 Smith et al. 2015 Ramirez et al. 2015 Kwon et al. 2017
B Fixation or immunofluorescence 	Prohemocytes  <10%	Oenocytoids  <10%	Granulocytes  ~80-95%	Castillo et al. 2006 Baton et al. 2009 Castillo et al. 2011
C Flow cytometry 	Prohemocytes n.d.	Oenocytoids n.d.	Phagocytic cells (granulocytes)  ~20-45%	This study Oliver et al. 2011

As displayed in the above figure, the percentage of granulocytes can vary greatly between the two approaches, with the increased percentage of granulocytes due to their increased adherence to a glass slide

following fixation. Prohemocytes and oenocytoids are non-adherent and washed off during the fixation process. The hemocytometer better accounts for all hemocyte subtypes present in the hemolymph. We recognize that these methods are not perfect, with these discrepancies also highlighting current limitations in our ability to distinguish hemocyte subtypes.

In our revised manuscript, we provide additional text in the results, methods, and figure legends to make these differences in the approach more transparent.

- 4. Since the authors observed an effect on virus prevalence in the gut (figure 1), they then bypass the influence of phagocyte depletion on midgut infection by injecting clodronate liposomes three days post-infection. While they looked at dissemination in different tissues, they did not check for the midgut infection. Flaviviruses takes a long time to disseminate from the gut, more than 3 days. In the absence of measure of infection in the midgut showing no difference in infection titers and prevalence, the conclusion that there is less dissemination after depletion cannot be supported.**

We appreciate the reviewer's comment. To clarify, we infected wild type mosquitoes with virus, then at three days post-infection, we randomly treated mosquitoes with control or clodronate liposomes to address dissemination. While we did not evaluate these mosquitoes for viral loads, there is no reason to believe that there would be any difference in the random distribution of virus infection in mosquitoes used for the dissemination experiments.

Moreover, based on the lack of effects of clodronate treatment on midgut viral load (Fig. 1) or on viral replication (Fig. S2), we do not feel that evaluating midgut titers in our dissemination experiments is required. With proper controls (the injection of "empty" control liposomes), we believe that our dissemination results in the context of phagocyte depletion via clodronate liposomes, can accurately be depicted as such.

- 5. Figure 4: titres, and not only prevalence, for all tissues should be shown.**

We now include viral loads and prevalence of infection for all tissues examined in a revised Figure 4.

- 6. When transferring supernatant or hemocytes (fig 5), the authors measure by qPCR the RNA viral loads in whole mosquitoes. First, this is not a measure of transmission to other tissues, could just be the initial infected hemocytes transferred. Hemocytes are known to divide, so the increase in virus load by qPCR seen with DENV may not be infection of mosquito tissues but an increase of hemocyte infection. Second, while viral loads measured by qPCR are more and more accepted although some will still not accept this as proof of infection (as opposed to virus titration, ie live virus), for this particular transfer experiment, I would have liked to see virus titration to make sure the virus transferred is still live and not just genome traces, especially in the case of ZIKV where only one time point was looked.**

Thank you for the suggestion. In our revised manuscript, we provide additional data demonstrating that virus can infect multiple tissues (salivary gland, ovary, midgut, and dissected carcass) post-transfer using qPCR (Fig. 5c), and further validate that this transfer of is a viable infection using focus-forming assays (FFAs) in select salivary gland and ovary tissues (Fig. S7).

- 7. The fact that virus collected from the supernatant fraction resulted in a significantly attenuated infection, and to be honest, it is not attenuated, two data points show clearly no infection at all, puts into question**

the data obtained with DENV, regarding the fact that cell free virus in hemolymph cannot participate to dissemination. Virus inactivation when not in a cell is rapid and the fact that the virus is inactivated after hemolymph perfusion, centrifugation does not necessarily mean the cell free virus does not contribute to dissemination within one mosquito.

Thank you for the comment. We believe that the lack of infectivity seen in our supernatant fraction is very interesting, yet at present, we don't have enough information to determine what is causing the reduced infectivity of virus present in the hemolymph fraction.

Our data suggest that there is equivalent amounts of DENV and ZIKV in the acellular supernatant and cellular fraction (Fig. 5e), yet for DENV there is reduced infectivity of the supernatant when used for infection of C6/36 cells *in vitro* (Fig. 5f) and *in vivo* through transfer experiments (Fig. 5b). This becomes more perplexing for ZIKV, where *in vitro* infection experiments display no difference between supernatant and cellular fractions (Fig. 5f), while comparable *in vivo* transfer experiments show an attenuated infection in mosquitoes challenged with the ZIKV supernatant fraction (Fig. 5b).

While we cannot rule out the potential that the isolation and manipulation of the supernatant fraction (perfusion, centrifugation, etc.) may alter the ability of both viruses to infect mosquitoes, the incomplete phenotypes observed in our *in vitro* and *in vivo* assays support that there also may be some unknown biological effect. For instance, it is known that DENV-infected cells release high levels of immature virus in mammals, yet lack the ability to infect cells. Therefore, it is unknown our collection of virus in the supernatant may similarly represent high levels of immature virus. In addition, other unknown hemolymph may also bind virus present in the hemolymph to quickly inactivate virus, preventing infection. As a result, additional studies beyond the scope of the manuscript are required to better understand these observed phenotypes.

In our revised manuscript, we have added additional text to our discussion to better address these possibilities and limitations of our current study.

- 8. I recommend the authors to stay prudent with conclusions about cell free virus and do not overstate that the hemocytes are the primary component of mosquito hemolymph that is able to promote virus dissemination. Hemolymph free virus may contribute, and this does not remove the interesting data that hemocytes also do. Regarding hemocytes, I'd like to see more controls, such as virus titration of different tissues, and not just whole mosquitoes. Otherwise, the conclusion that granulocytes are capable of disseminating viruses to mosquito tissues (line 203-204 for example) is invalid.**

Similar to the above comment, we have added additional text in the discussion of our revised manuscript regarding the potential that our manipulation of virus in the hemolymph through perfusion and centrifugation may alter its infectiousness.

In addition, as previously stated in one of the above comments, we provide additional experiments in our revised manuscript that demonstrate that virus-infected hemocytes transferred to a naïve host can infect multiple tissues (salivary gland, ovary, midgut, and dissected carcass; Figure 5c), and further validate that this transfer of is a viable infection using focus-forming assays (FFAs) in select salivary gland and ovary tissues (Figure S7). We believe that these new experiments should satisfy the desire for additional rigor in our manuscript.

9. L94: CLD treatment may still be efficient thanks to liposome presence after 10 days. In addition, the study looks up to 10 days, therefore, I would recommend staying cautious with the statement that “phagocyte depletion is permanent for the life of the mosquito”, which can live up to a month or so. Rephrase, eg phagocyte depletion was efficient at least up to 10 days.

Thank you for the suggestion. This has been corrected to “phagocyte depletion is efficient for at least 10 days” in our revised manuscript.

10. L100: viral titers by qRT-PCR > viral loads (titres are obtained with titration, not qPCR) – To check all manuscript including legends.

Thank you for the suggestion. We have changed any use of “titers” to viral loads (or something similar) in our revised manuscript.

11. References: Virus infected by arboviruses (alphavirus ONNV) and hemocytes attached to tissues, including midgut and trachea, please cite Hemocyte-targeted gene expression in the female malaria mosquito using the hemolectin promoter from *Drosophila* – ScienceDirect

The study by Pondeville *et al.* has been added to our revised manuscript to support instance of hemocyte infection by arboviruses and attachment to mosquito tissues as requested. We apologize for the oversight.

Reviewer #2

This manuscript demonstrates the role that hemocytes play in the dissemination of two medically important flaviviruses within the hemocoel of the mosquito vector. The data are significant because hemocytes are considered protective against infection, yet the demonstration here is that they can also be a liability, presumably affecting vectorial capacity. Overall, I am supportive of the manuscript but below highlight some important points.

We would like to thank the reviewer for their positive comments regarding the importance of our data, as well as the helpful comments to improve our manuscript. We have addressed each of the specific comments below.

Reviewer Comments

1. Granulocyte percentage and identification in figures S1 and 2. Figure S1 shows that CLD treatment decreases the percentage of granulocytes, and places granulocytes as 8-15% of the hemocyte population. Yet, in figure 2a the granulocytes are prevalent in the image; 8 of the 9 cells (89%) are phagocytic and presumably granulocytes. In figure 2b, the percentage of granulocytes is quantified as 90% or greater (add the first two columns). How can this contradiction be reconciled? Also, can the cell types in figure 2a be labeled?

As mentioned already in our comments to R1 (Comment #3), this discrepancy is due to the different previously published techniques used to determine the percentage of granulocytes. Briefly, the data presented in Fig. S1 were performed using hemocytometer assays, while those in Fig. 2 were performed via immunofluorescence using fixed hemocytes on a slide. In our revised manuscript, we have made changes to make these differences in the manner that they were examined more transparent.

- 2. Figure 3. Frequency of occurrence matters. Figure 3 demonstrates that hemocytes can attach to the salivary glands and ovaries, which is relevant to the scientific question because these are important destinations of the virus. However, the frequency of these occurrences is not mentioned. Are hemocytes most often seen attached to these tissues, or do they attach seldomly? Are there usually many hemocytes at these locations, or are they usually few?**

Thank you for the suggestion. In our revised manuscript, we examine hemocyte attachment to the salivary glands, ovaries, and midgut under naïve, blood-fed, and DENV-infected conditions at multiple timepoints. These new data are integrated into a revised Fig. 3 in our revised manuscript.

- 3. Figure 4. Magnitude matters. The findings in this figure are presented in binary terms. Treatment either affects or does not affect. More nuance is needed. For example, in figure 4b all timepoints are significantly different, but the differences at the later two timepoints are small. Likewise, the prevalence phenotypes in figure 4C are significantly different in some timepoints but not in others. For transparency, the narrative should capture the nuances.**

In our revised manuscript, we now include virus copy numbers and infection prevalence for DENV and ZIKV for the legs, ovaries, and salivary glands for each of the timepoints examined in a revised Fig. 4. In addition, we have added additional text to the results and discussion of our revised manuscript to enhance the narrative and the description of these dissemination data in a larger biological context.

- 4. Hemolymph transfusion experiments, part 1 (Fig 5a-c; start in line 162). These experiments are very interesting, and difficult. In reading the manuscript, I had difficulty conceptualizing some of the findings, primarily because the hemolymph being used is collected via perfusion. In other words, it is not pure hemolymph, and in fact, the majority of the fluid is the transfer buffer and not hemolymph. More details on the methodology are needed. Hemolymph is estimated to be what percentage of the collection? Is the CELL component more concentrated than the SUP component? Was the cell component microscopically visualized to see whether granulocytes are there? How can it be explained that viral titer in CELL and SUP are similar yet there are some differences in how they infect cells in vitro?**

Thank you for the suggestion. In our revised manuscript we have added additional experimental details in our methodology to address the above comments as best as possible to enhance the rigor and reproducibility of our experiments. We have also added to the discussion some potential reasons as to why we observe differences in the supernatant and cellular fractions. However, additional experiments, beyond the scope of our manuscript, are required to better understand any differences in infection between the supernatant and cellular fractions.

- 5. Hemolymph transfusion experiments, part 2 (Fig 5d-f; start in line 192). Could it be that the lower infectivity when hemolymph from CLD mosquitoes is transfused is simply because dissemination has been lower (see fig 3).**

As the reviewer points out, we cannot rule out that the amount of virus in the hemolymph of CLD-treated mosquitoes may be lower than LP controls since we did not quantify the amount of virus present in these samples. While this may have some influence on the interpretation of our experiments (now Figure 5h of our revised manuscript), we would argue that these experiments are more to validate the observations obtained in wild-type mosquitoes (Figure 5a-c), which suggest that the cellular fraction is more efficient at the transfer

of DENV and ZIKV. Therefore, we consider these experiments to be more confirmatory experiments that demonstrate that phagocytic granulocytes (targets of clodronate depletion) are the cell types implicated in transfer of virus to naïve mosquitoes.

- 6. In the discussion, the authors tackle two potential discrepancies between their data and the data of others. The discrepancy outlined starting in line 215 makes complete sense, and I agree with the authors (I would also expect phagocytosis saturation and CLD treatment to yield different outcomes). However, I am less convinced regarding the discrepancy that starts in line 230. Cheng et al 2022 claim that prohemocytes are the virus infected cell whereas the present manuscript states that granulocytes are the virus infected cell. I do not know the answer, and I see problems with the claims in both studies. As pertains to the manuscript being evaluated here, the authors demonstrate that granulocytes are infected, but there isn't any convincing evidence that prohemocytes are not. Given that Fig S1 shows that 8-15% of the hemocyte population are granulocytes and fig 2 focuses only on granulocytes, even a small percentage of prohemocytes (presumably most of the remaining 85-92%) infected could dwarf granulocyte infection.**

Thank you for the opinion and insight regarding these discrepancies between our work and other previously published experiments. We are also appreciative of the reviewer's comment regarding potential roles of prohemocytes in virus infection, and have amended the discussion of our revised manuscript to explore this possibility in full transparency of the limitations of our study. While we believe that granulocytes are the predominant hemocyte subtype involved in virus infection, we agree that we can't fully eliminate the possibility that other cell types may also become infected. This topic is mentioned in the discussion of our revised manuscript.

- 7. It would have been highly informative to see data on mosquito survival. Does CLD treatment cause more mosquitoes to die when infected with a virus? If so, this would suggest that hemocytes enhance dissemination while protecting the mosquito's reproductive success. In other words, from the perspective of the mosquito, the hemocytes would still be beneficial.**

In our revision we have included survival experiments that examine the effects of clodronate treatment on mosquito survival (control liposomes vs clodronate liposomes) under blood-fed and virus-infected conditions. For the results provided in Fig. S3, we see no effect of treatment or infection status.

- 8. One final thought that the authors should consider is that the importance of granulocyte infection could be more so as replication factories and less so as dissemination vehicles. This does not challenge anything in the manuscript, but I mention it because the dynamic nature of hemolymph flow might mean that, inside hemocytes, replication may be more important than transportation.**

Thank you for the suggestion. There is definitely the potential that virus-infected hemocytes enable further replication of the virus, which subsequently enhances virus infection. We have amended the discussion of our revised manuscript to reflect this possibility.

Reviewer #3

The article by Hall et al addresses an important question about the role of circulating immune cells during arbovirus infection in mosquitoes. The major hypothesis in the manuscript is that mosquito phagocytes acquire the arboviruses in the mosquito gut and then are essential to disseminate these viruses to other tissues, such as the salivary gland. This is an interesting hypothesis that has been proposed before, but the article does not

present experiments that directly prove it. Basically, the manuscript has experiments that establish a correlation. First, authors show that hemocytes are infected by dengue and Zika viruses and find infected hemocytes adhering to the ovaries and salivary glands. However, this is a correlation and not proof that these hemocytes were the vessel for virus dissemination. It is possible that hemocytes might have migrated later to assist in the immune response against infection or to deal with tissue damage. Second, authors show that transference of infected hemocytes is very efficient at infecting new mosquitoes but not cell-free viruses. This is the highlight of this manuscript since most of the results presented are not new. For example, the pro-viral role of hemocytes during dengue and Zika virus infections has been demonstrated (PMIDs: 34093550) as well as the role of hemocytes as a target for arbovirus infections (PMIDs: 34093550, 25548172, 17263893, 19141437). However, the infectiousness of cell associated virus does not prove that hemocytes themselves are the vessel for virus dissemination from the mosquito gut to other tissues. I think this observation could be the basis for an entire manuscript to be developed but, at the moment, it lacks mechanistic understanding. Hence, I think this manuscript would be a better fit for a more specialized journal.

We appreciate the reviewer's summary and criticisms of our work. While we are aware that several studies have shown that virus can infect mosquito hemocytes, only a single study by Leite *et al* (PMIDs: 34093550) has provided any functional information as to how mosquito hemocytes contribute to virus infection. Although there are some areas of overlap between this previous study and our own work examining the role of hemocytes in midgut infection (Fig. 1 of our study), we believe that our study provides significant new insight into the importance of hemocytes in arbovirus infection, including the attachment to relevant mosquito tissues required for transmission, virus dissemination, and the role of hemocytes vs free virus in transferring virus infection. Moreover, our experiments rely on newly developed tools for phagocytic hemocyte (granulocyte depletion) that differ from previously published approaches and believe that the outcomes of our experiments represent a significant advance in our understanding of mosquito-virus interactions.

With the additional experiments that have now been added to our revised manuscript as a result of reviewer comments, we have improved the rigor and transparency of our data and their conclusions, such that we believe that our revised manuscript provides a strong case for the role of mosquito hemocytes in arbovirus dissemination. While we believe these experiments are at the forefront of mosquito hemocyte biology, we acknowledge that the lack of hemocyte markers and genetic tools currently limit our ability to study mosquito hemocytes through more definitive methods.

As the reviewer points out, we also believe that our study provides the foundation for novel investigations of the infectivity of cell-free and hemocyte-associated virus that have previously been unexplored, and agree that this observation is deserving of further study. However, we believe that this will take significant time to address and is well beyond the scope of our present manuscript.

We are appreciative of the reviewer's suggestions to improve our manuscript and have addressed each of these specific comments below.

Reviewer Comments

- 1. In figure 1 there is a clear trend of lower infection in the midgut when hemocytes are depleted, as previously reported when phagocytosis is blocked (PMIDs: 34093550). This is an important point that needs to be addressed, especially since the authors use clodronate in this manuscript compared to latex beads in Leite *et al* (PMIDs: 34093550). Based on the supplementary data, the experiment was performed three times. Does the figure show all of them combined? How do each of them separately look like? Overall, the authors have a model that suggests saturating infection, with over 80% of infected mosquitoes, which could mask**

any effects of hemocytes in the midgut. Thus I disagree that there is no effect on midgut infection when hemocytes are depleted. Authors could have looked at earlier time points and analyzed direct midgut infection by imaging as they did for other organs.

In our original submission, the data presented in Fig. 1 displaying midgut infection following clodronate treatment were displayed as pooled data from three or more replicates. With also a similar query from R1 regarding the replicates, in our revised manuscript we now display all of the individual midgut infection experiments for DENV and ZIKV as part of Fig. S2.

For our dengue infections, we had originally pooled data from four experiments, yet after further analysis suggested by R1, one of these experiments (Exp #2) was statistically different from the other LP control groups (via Kruskal-Wallis and Dunn's multiple comparison test). As a result, we performed an additional experiment (Exp #5) that did not display significant differences with the other control group data from individual experiments. We then pooled these infection data (copy number and prevalence), excluding Exp #2, as part of a revised Figure 1b. However, data for all five of the individual experiments are included in Fig. S2, with Exp #2 highlighted that it is not included in the pooled infection data. While there are differences between each of these five experiments, we do not see a significant effect of phagocyte depletion on DENV copy number or prevalence in either the pooled data or individual experiments. Therefore, these data suggest that phagocytes have no effect on DENV midgut infection.

The ZIKV midgut infection data remains unchanged in our revised manuscript, with the original pooled data displayed in Fig. 1c and the individual experiments (copy number and prevalence) displayed in Fig. S2. While we do not see a difference in ZIKV copy number, there is a significant difference in the prevalence of infection, with these trends consistent across each of the individual experiments. These data infer that phagocyte depletion does interfere with ZIKV midgut infection outcomes.

As the reviewer is likely aware, It can be difficult to control the infection prevalence in oral infection experiments. For both our DENV and ZIKV infections, we see ~70% infection prevalence with individual experiments/conditions ranging from ~30-100% prevalence. While we can't rule out the potential that feeding with a lower amount of virus may resolve additional infection phenotypes, we disagree that the data presented are of a saturating infection. The fact that we see significant differences in the prevalence of ZIKV infection of the midgut supports this statement.

Although not included in our manuscript, we have performed preliminary experiments that examine ZIKV infection of the midgut at earlier timepoints (3 dpi) as shown on the left. In these initial experiments, we did not see significant differences in ZIKV copies or infection prevalence, such that we did not explore these experiments further. Moreover, these data suggest that we did not overlook any infection phenotypes that may occur at earlier timepoints in infection. Due to the preliminary nature of these experiments, these data were not included in our revised manuscript.

In our opinion, we believe that the qRT-PCR methodology to determine infection outcomes provides an increased sensitivity and is more easily quantified than the presence/absence of virus through immunofluorescence (IFA) experiments as suggested by the reviewer. For this reason and the lack of a discernable phenotype at these earlier timepoints, we opted not to further examine midgut infection by IFA as in Leite et al.

- 2. Figure 2 shows data for 10 days post infection – how representative would this be during the kinetics of the infection? According to the major hypothesis, hemocytes should be infected early. Here it would also be good to use the same dye and in figure 3, since beads could affect hemocyte physiology.**

We would like to thank the reviewer for the suggestion. In our revised manuscript, we provide additional data that examines the kinetics of hemocyte infection. In these experiments, now included as part of a revised Fig. 2, we examine hemocyte infection every 2 days over a 10-day period. We see hemocyte infection with DENV beginning at ~6 dpi, suggesting that hemocytes become infected only after virus has successfully undergone replication in the gut prior to dissemination. This new information has been incorporated into a revised results and discussion section to reflect these findings and to update our model of hemocyte infection.

- 3. In figure 3, in order to have more meaningful information, the number of hemocytes attached to salivary glands and ovaries should be quantified during the kinetics of viral infection. This should be compared to the infection in these organs as compared to the recruitment of hemocytes to establish the idea that hemocytes arrive before the infection becomes clear. The presence of hemocytes in these organs should also be analyzed in the absence of infection. It has been shown that injection of beads can increase the amount of hemocytes associated to the gut (PMIDs: 34093550) and this may be true for other tissues. Thus, it is important to have a control group without beads.**

We would like to thank the reviewer for the suggestion. In our revised manuscript, we have examined the effects of bead injection on hemocyte attachment to the salivary glands, ovary, and midgut, demonstrating that bead injections increase hemocyte attachment to each of the respective tissues similar to that previously described in Leite *et al.* We also examine hemocyte attachment in the absence of beads to the salivary glands, ovary, and midgut under different physiological conditions (naïve, blood-fed, and DENV infection), providing data that suggests that hemocytes bind to each tissue independent of infection status. These data have been incorporated into Figure 3 of our revised manuscript.

- 4. In figure 4, viral titers should be shown for all results. Delaying dissemination or directly hosting viral replication would basically translate into the same effect and the authors cannot exclude the two possibilities. Furthermore, without knowing whether there is a clear effect on midgut infection, it is hard to analyze anything after, since dissemination depends on infection at the primary site.**

We now display DENV and ZIKV infection data for each of the respective tissues/timepoints as requested in a modified version of Figure 4 in our revised manuscript.

The reviewer does make a strong point that the delays in virus dissemination following phagocyte depletion can also be caused by the reduced virus replication in the hemocyte tropism. R2 made a similar comment and have modified the discussion of our revised manuscript to include the potential that the observed phenotype may be due to the loss of secondary virus amplification in hemocytes.

We also believe that we have adequately addressed questions of phagocyte depletion on midgut infection. The data provided in Figure 1 suggest that there is no effect on DENV or ZIKV copy numbers, although there is an effect on ZIKV infection prevalence. We believe that the inclusion of the individual experiment data in our revised Figure S2 provide better transparency of these results. In addition, we believe that the inclusion of *in vitro* virus infection data in the presence of clodronate disodium salt (Figure 1d-f), which has no effect on virus replication, provide additional support for the specificity of our methods to assess virus infection in the context of phagocyte depletion. Moreover, we believe that our experimental approach to address questions of dissemination in which we performed clodronate treatment at 3 dpi, minimize any potential differences that phagocyte depletion may have on midgut infection. Therefore, we believe that our experiments provide the best approach to examine virus dissemination in the mosquito host.

- 5. The results in figure 5 are the highlight of the manuscript, but they do not prove the authors hypothesis. Results that cell associated viruses are more infectious to new mosquitoes do not prove that viruses are disseminated within hemocytes in the infected mosquito. Furthermore, authors show that dengue in the supernatant is less viable than in cell fractions, although not true for Zika, which deserves further analysis. For example, is it a matter of timing? If they were to analyze an earlier time point with Zika, would it be different? Overall, the observation is poorly explored by the authors. Supplementary figure 3, that complements these results, should be integrated into a main figure. More experiments are required to help explain these results.**

We would like to thank the reviewer for their comment, yet respectfully disagree. We believe that our results showing that the transfer of the cell fraction results in the transfer of a virus infection to a naïve host (Figure 5a-b), that when paired with new data demonstrating that cell transfer promotes virus infection of salivary gland, ovary, midgut, and carcass tissues (Figure 5c; Figure S7), provide strong support for the role of hemocytes in virus dissemination.

As the reviewer points out, we agree that the data with the supernatant are interesting and are highly deserving of further study. From the data collected in our study, these data support that the supernatant fraction is attenuated or less infectious in transfer studies to naïve mosquitoes despite equivalent viral loads (now Figure 5e). While the supernatant fraction for DENV displays limited ability to infect C6/36 cells, the ZIKV supernatant fraction infect with equal ability to that of the cell fraction (now Figure 5f). As a result, it is unclear how timing, the presence of immature virus, or inactivating factors present in the mosquito hemolymph may contribute to these observed phenotypes for both viruses. While we discuss these possibilities in greater detail in our revised manuscript, answering these questions will be a significant investment in time and effort, which in our opinion, extends well beyond the scope of our current manuscript.

As suggested, supported data previously included as Figures S2 and S3, which examine viral copy numbers and infection of C6/36 cells in the supernatant and cell fractions, have now been included in as Figure 5e-f in our revised manuscript.

- 6. In figure S2, statistics on 2 replicates should be removed.**

We have removed the statistical analysis for what is now Figure 5e in our revised manuscript.

Response to Reviewer's comments

NCOMMS-24-24225A

"Mosquito immune cells enhance dengue and Zika virus infection in *Aedes aegypti*"

A detailed response to each of the reviewer comments is listed below. All changes in response to the reviewer's comments are highlighted in the manuscript text.

Reviewer #1

Every point raised has been addressed. Congratulations to the authors for this excellent paper!

Thank you for the kind words. Much appreciated and thank you for your efforts.

Reviewer #2

This is an improved manuscript demonstrating that dengue and Zika viruses can replicate in granulocytes to amplify the infection, and that these infected granulocytes can be a vehicle for the artificial transmission of viruses from one mosquito to another. The body of work is impressive, and the experiments are well designed and conducted. The findings are significant, but my main concern is the overinterpretation of the meaning of the data. I will focus on two main points:

- 1. In the rebuttal, the investigators explain that the difference in the cell type percentages between figure panels (granulocytes versus prohemocytes versus oenocytoids) is that different methods yield different outcomes: using a hemocytometer, granulocytes are the smallest population, but using microscope slides they are the largest population. The claim is made that the hemocytometer provides the true proportion. The experiment where the authors conclude that granulocytes are the main cells that becomes infected used the microscope slide method. Using this method, the authors note that 70-95% of cells are lost (most of the prohemocytes and oenocytoids), so without the other cells present it seems likely that granulocytes would be the most infected. Therefore, can it really be claimed that granulocytes are the predominant infected cell when the method dramatically enriches this population? This is especially the case because it is not noted whether prohemocytes (which have some phagocytic activity) or oenocytoids are depleted by clodronate. For that reason, an unsupported claim is that "These results provide strong evidence that phagocytic granulocytes comprise the majority of virus-infected hemocytes and indicate that phagocytic granulocyte populations may be important for virus dissemination".**

Thank you for the suggestion. While we are confident that granulocytes are likely the predominant infected cell type, we do recognize the possibility that other hemocyte populations could be infected by virus as suggested by Cheng *et. al.* 2022. However, this is a challenging question to further examine given the current technical limitations in the study of mosquito hemocytes where different methodologies yield different immune cell percentages and a lack of cell type-specific markers to distinguish hemocyte subtypes. While this is a question that we hope to further resolve in the future, we acknowledge these current limitations. As a result, we have modified the aforementioned claim in our revised manuscript to the following:

"These results suggest that the majority of virus-infected hemocytes are phagocytic. However, due to technical limitations in our ability to define mosquito hemocyte subtypes, our data can only infer that the predominant immune cells involved in virus infection are phagocytic granulocytes, and cannot fully exclude the potential roles of other immune cell populations in virus infection."

- 2. Another primary claim is that hemocytes disseminate the infection. A clear distinction between replication and dissemination should be made. Replication means that viral copy number increases, and the manuscript makes a very convincing argument that this is the case. Dissemination means that the virus is transported, and this claim is significantly weaker. Hemocytes attach to all tissues, and surely their attachment to the salivary glands brings the virus closer to where they need to be for transmission. But this does not discount the possibility that soluble virus is infecting the salivary glands (or other tissues). The claim for dissemination is made using two lines of evidence. The first is that CLD treatment reduces virus dissemination to the legs. The experiment demonstrates that there is a delay in dissemination but not that the hemocytes are disseminating (other than by being replication factories). The second is that injection of the cellular portion of hemolymph causes infection in naïve mosquitoes but injection of cell free hemolymph does not. The problem here is that the cellular fraction is concentrated by centrifugation (so it may be more concentrated than actual hemolymph) but the cell free fraction is significantly diluted by the solution used to perfuse. So, the experimental design shows that infected hemocytes can infect, but the statement that the cell free fraction is not infective is weak.**

Thank you for the comment. To the *first point* regarding virus replication/dissemination, we believe that our data strongly implicate mosquito hemocytes in the spread of virus infection to peripheral mosquito tissues. We demonstrate: **1)** that phagocytic immune cells are infected by virus; **2)** that phagocytic hemocytes attach to peripheral mosquito tissues (salivary glands and ovaries); **3)** that phagocyte depletion impairs virus titers/infection prevalence to the legs/salivary glands/ovaries; and **4)** that transfer of virus-infected phagocytic hemocytes are able to confer a virus infection to naive mosquito host tissues. While we concede that at this time we do not provide definitive roles of hemocytes in either virus replication or dissemination, in our opinion, there is little doubt for their involvement in the spread of virus infection. As a result, we have tempered our claims that hemocytes are involved in “dissemination” throughout our manuscript (title/abstract/results/discussion), changing the wording from “dissemination” to a more simplistic and all-encompassing term such as “infection”. We believe that this should provide a more grounded role in the current interpretation of our data, that we hope to build upon in future experiments to better define the role of hemocytes in replication and dissemination.

To the *second point*, we disagree that the cell-free supernatant fraction was more dilute than our cell fraction. Both the supernatant and hemolymph fractions were equally diluted during the handling of the perfused samples in our experiments, and while they may not be as concentrated as each fraction may be in vivo, should indicate the infectiousness of the respective fractions. As stated in our methods, after the cell fraction was collected by centrifugation, the sample was diluted with an equivalent volume to that of the supernatant fraction. In theory, this should normalize each fraction into relatively equal “mosquito equivalents” for each supernatant or cell fraction, for which the same volume was transferred to recipient mosquitoes.

Our data displayed in Figure 5b and 5h suggest that there are significant differences in the ability of the cell-free and hemocyte fraction to infect naïve mosquitoes, yet we are careful to state that the cell-free supernatant fraction is “less efficient”, rather than unable to infect. We are very aware of the limitations of these experiments and discuss these at length in the discussion of our manuscript. These data are even more interesting given that these fractions contain biologically similar amounts of virus (Figure 5e), which also support that one fractions is not more concentrated than the other.

With additional data displaying differences for DENV and ZIKV in their ability of the cell-free supernatant and cell samples to infect C6/36 cells (Figure 5f), we believe that there is likely something biologically that is

shaping these differences in virus infection. We find the outcomes of these experiments to be very intriguing and believe that they impart the need to further explore these results (as similarly suggested by R3) through additional experiments. However, given the experimental challenges with these perfused hemolymph samples, the relative unknowns in tackling these questions, and amount of time required to conclusively address these questions, we believe that these experiments are well beyond the scope of this manuscript.

Reviewer #3

The revised article by Hall et al titled "Mosquito immune cells enhance dengue and Zika virus dissemination in *Aedes aegypti*" has improved over the previous version but some key questions remain. I appreciate the efforts from the authors but, overall, the data do not fully support the hypothesis that hemocytes work as trojan horses to disseminate dengue and Zika viruses. Without further mechanistic insights and significant new data, I think the manuscript is more appropriate for a specialized journal. Below are my major criticisms:

1. The data on the kinetics of hemocyte infection (Figure 2), hemocyte association with tissues (Figure 3) and tissue infection (Figure 4) together show a scenario that does not fit the idea that hemocytes are themselves the vehicle for virus dissemination. First, hemocytes clearly become infected at the same time or even later than other tissues. Second, how do the authors address the matter that hemocyte association with tissues is not affected by the infection? How do hemocytes help spread the virus if they are always associated with the tissue? If they are already associated with tissues such as ovaries and salivary glands at 3 days post feeding to the same levels but only become infected at 6 dpf, how do the authors suggest they spread the virus. This would require a lot of re-localization of hemocytes. Authors should show infected hemocytes in the tissue and not only cells obtained after perfusion. Third, the levels of tissue associated hemocytes does not match the idea they drive virus dissemination. There are at least 5 times more hemocytes associated with ovaries than salivary glands in their dataset but, yet, infection is higher in salivary glands. How to account for these inconsistencies?

Thank you for the comments. In our revised manuscript, we have tempered our arguments stating that hemocytes are involved in virus dissemination. As mentioned above, Reviewer #2 had similar reservations regarding our previous submission, and as a result, have made significant alterations to our revised manuscript, referring to the roles of hemocytes in virus "infection" as opposed to virus "dissemination". This includes changes to the title, abstract, and throughout our revision.

Regarding *your specific comments above*, we address each individually below:

First, hemocytes clearly become infected at the same time or even later than other tissues.

Our provided data suggest that hemocytes become infected at ~6dpi. This is an approximate time point (we only measured every 48 hrs), where the timing may slightly vary between individual mosquitoes based on the intensity of virus infection. Of note, we believe that this corresponds to the approximate timing of when DENV/ZIKV begins to escape the midgut and subsequently spread to other tissues based on previous publications (PMIDs: 8834747, 17263893, 38557892). While our data support the involvement of hemocytes (particularly phagocytic hemocytes) in the infection of the legs, salivary glands, and ovaries; these phenotypes are not absolute, and we believe that we are transparent in that there may be other routes for the spread of virus in the mosquito host.

It is worth noting that hemocyte infection begins at day 6 (Figure 2C), yet we only evaluate the phenotypes of these peripheral tissues at 8, 10, and 12 dpi. Our most significant phenotypes are consistently at the 8 and 10 dpi timepoints for both DENV and ZIKV.

Second, how do the authors address the matter that hemocyte association with tissues is not affected by the infection? How do hemocytes help spread the virus if they are always associated with the tissue? If they are already associated with tissues such as ovaries and salivary glands at 3 days post feeding to the same levels but only become infected at 6 dpf, how do the authors suggest they spread the virus. This would require a lot of re-localization of hemocytes. Authors should show infected hemocytes in the tissue and not only cells obtained after perfusion.

In our hands, we did not see significant differences in hemocyte attachment to tissues in response to blood-feeding and virus infection, which contrasts previous observations by Leite et al. (PMID: 34093550). While we do not discredit these contrasting observations, we do not believe that hemocyte attachment is permanent and is thus a transient stage. As a result, we view these results as a snap shot of hemocyte attachment at any given time, which is an unfortunate endpoint for each individual mosquito at a given time and physiological status. There are a few studies supporting that hemocytes can attach to tissues, then re-enter hemolymph circulation (PMID: 18632567, 26526332), which have now been included in the discussion of our revised manuscript.

To the reviewer's point of examining infected hemocytes attached to various tissues, it is possible to examine infected hemocytes attached to mosquito tissues with further experimentation, yet we believe that this only provides additional circumstantial evidence. Given the technical challenges associated with these experiments and the amount of time that would be needed to examine multiple tissues and timepoints for little gain beyond our existing quantification of virus infection, hemocyte attachment, phagocyte depletion, and transfer experiments that already clearly implicate roles for hemocytes in the spread of virus infection. As a result, we believe that these types of experiments are beyond the scope of our current manuscript.

We would also like to remind the reviewer that many questions still remain regarding the roles of hemocytes in virus infection. Given the scant number of previous papers that have approached these questions, we believe that our manuscript provides a strong foundation to further explore hemocyte-virus interactions. However, it is unreasonable to think that we provide every answer to this challenging and understudied tropism for virus infection solely within this manuscript. We look forward to examining a wide-array of questions regarding hemocyte-virus interactions in future experiments.

Third, the levels of tissue associated hemocytes does not match the idea they drive virus dissemination. There are at least 5 times more hemocytes associated with ovaries than salivary glands in their dataset but, yet, infection is higher in salivary glands. How to account for these inconsistencies?

We believe that the raw numbers of hemocytes associated with each tissue is relative to size. While the midgut and salivary glands are not influenced by physiological status, the ovaries dramatically increase in surface volume following vitellogenesis. As mentioned in our manuscript, we believe that this physical difference in size accounts for these differences in raw numbers. Simply put, the bigger the tissue, the more surface area are allowing for greater hemocyte attachment.

While we do not explore ways to normalize these raw hemocyte counts to tissue volume, we hypothesize that the salivary glands likely have the highest attachment of hemocytes relative to tissue size.

- 2. I reinforce that the preliminary data shown in Figure 5 are the highlight of the manuscript that can be developed into something interesting. However, authors have not addressed my previous comments and no new further data was included.**

We appreciate the sentiment, but as we explained in our previous submission, we strongly feel that to provide any type of resolution to these questions is not trivial and would potentially require years of work to address. In our opinion, this significant investment of time, energy, and resources does nothing to change (only potentially enhance) the primary conclusions of our study that hemocytes help promote virus infection.

Similar to our response to R2, given the experimental challenges with these perfused hemolymph samples, the relative unknowns in tackling these questions, and amount of time required to conclusively address these questions, we believe that these experiments are well beyond the scope of this manuscript.

If the reviewer has any constructive suggestions to address these outstanding questions in a conclusive and timely fashion, we would be happy to entertain these experiments.

- 3. The experiment described in Figure 4 shows a marginal effect on viral loads measured by RT-qPCR but a significant effect on prevalence. However, that is not reproduced when the authors look at virus titrations (Supplementary Figure S6). What is their explanation?**

The results presented in Figure S6 using FFA are meant to address previous reviewer comments regarding the infectiousness/viability of virus in the associated tissues as a comparison to the use of qRT-PCR experiments that quantify viral copy numbers. In these experiments, we demonstrate that there is a productive infection, yet the number of tissues examined in these experiments was limited with ~10 individuals examined for each treatment/tissue. Even in those experiments that display promising results, we admittedly still lack the statistical power to come to any firm conclusions, nor do we attempt to oversell these experiments.

These efforts took significant time and energy to complete, and in my opinion were a valuable addition to our study. However, we would have to examine 40-50 additional mosquito tissues and treatments to examine these data with similar statistical power to the already provided qRT-PCR quantification.

Frankly, I am not aware of any study that has provided that level of redundancy and rigor, especially when qRT-PCR quantification is already a widely accepted methodology to determine the outcomes of virus infection. We view the provided qRT-PCR data as the primary means of evaluation, with the FFA serving only to supplement and not completely reiterate these analyses.

- 4. RT-qPCR to detect virus in RNA extracted from dissected tissues does not show active replication in the tissue as the authors mention in their rebuttal. In contrast, IFA of the tissue is much better at determining where the virus is replicating.**

Similar to the above question, in our previous revision we did examine virus infection levels (active infection) using FFAs. While we concede that these were not performed in a large number of individuals, they more than illustrate an active and productive virus infection in these tissues.

Moreover, while IFAs may provide a nice visual confirmation, we argue that the FFAs provide a more exact quantification of virus loads than that of an image. Even with imaging software, pixel intensity doesn't fully equate to the actual quantification of virus and can be artificially manipulated during processing.

Response to Reviewer's comments

NCOMMS-24-24225B

"Mosquito immune cells enhance dengue and Zika virus infection in *Aedes aegypti*"

A detailed response to the editorial and reviewer comments are listed below. All changes in response to the reviewer's comments are highlighted in the manuscript text of a separate marked file.

Reviewer #2

The authors have addressed my comments and I am supportive of the manuscript.

We would like to thank the reviewer again for their time and efforts in helping to improve our manuscript.

Reviewer #3

This newly revised manuscript by Hall et al did not provide any new compelling evidence to change my opinion about the work. I appreciate the carefully crafted response but, I have to repeat myself that without further mechanistic insights and significant new data, I think the manuscript is more appropriate for a specialized journal. On many of the specific points, I agree with the authors but they did not address the major concerns. It is true that there is not a lot of information about the contribution of hemocytes to arbovirus infections in mosquitoes. While this means that many questions remain unanswered, the work by Hall et al did not bring a significant contribution to the subject. The message that hemocytes have a proviral role during dengue and Zika virus infections is not novel. I reinforce that data in figure 5 should be the main focus of the work but it is just an initial basis for the hypothesis that hemocytes are the vessels for virus dissemination. I appreciate the response from the authors but it is not enough, more substantive data is required to support the hypothesis.

We would like to thank the reviewer again for their time and efforts in helping to improve our manuscript. We believe that the data in our manuscript provide an important foundation for our continuing work to better understand the roles of hemocytes in virus infection, and hope that in the future that we may be able to provide the mechanistic insights desired by the reviewer.